# Peptide-enhanced tough, resilient and adhesive eutectogels for highly reliable strain/pressure sensing under extreme conditions

Yan Zhang[1], Yafei Wang[1], Ying Guan [1]✉ & Yongjun Zhang [2]✉

Natural gels and biomimetic hydrogel materials have been able to achieve outstanding integrated mechanical properties due to the gain of natural biological structures. However, nearly every natural biological structure relies on water as solvents or carriers, which limits the possibility in extreme conditions, such as sub-zero temperatures and long-term application. Here, peptide-enhanced eutectic gels were synthesized by introducing α-helical "molecular spring" structure into deep eutectic solvent. The gel takes full advantage of the α-helical structure, achieving high tensile/compression, good resilience, superior fracture toughness, excellent fatigue resistance and strong adhesion, while it also inherits the benefits of the deep eutectic solvent and solves the problems of solvent volatilization and freezing. This enables unprecedentedly long and stable sensing of human motion or mechanical movement. The electrical signal shows almost no drift even after 10,000 deformations for 29 hours or in the −20 °C to 80 °C temperature range.

Flexible strain/pressure sensors, mimicking human skin by detecting external stimuli and converting them into electrical signals, allow monitoring human physiological information in a noninvasive way, which are also promising in soft robotics, human-machine interfaces, etc[1]. Conductive hydrogels, a kind of traditional soft conductive materials, have been extensively investigated for this purpose. Compared with other soft conductive materials such as filler/elastomer composites, conductive hydrogels exhibit advantages including excellent biocompatibility, tunable mechanical properties, relatively high electrical conductivity, and low interfacial resistance. Therefore they have attracted more and more attention in recent years[2].

Significant progress has been made in the development of hydrogel-based flexible sensors, however, the reliability and stability of the sensors, particularly in long-term use and under extreme conditions, still remains a tremendous challenge[3–5]. To achieve this goal, a conductive hydrogel should meet a wide range of requirements

simultaneously. First of all, the gel should be stretchable and tough enough. Unfortunately ordinary synthetic hydrogels are quite fragile because of their inhomogeneous network and the lack of mechanism for energy dissipation[6]. Their strength should be enhanced using strategies, such as double network gels[7–9]. nanocomposite gels[10], and dual-cross-linked gels[11]. Secondly, the gels should be highly resilient to achieve reliable and stable response. Unfortunately for gel materials, high resilience and high toughness are seemingly contradictory[12]. High strength hydrogels usually have a low resilience[13] leading to low signal reproducibility and severe baseline drift of the sensors[7,14–17]. A third requirement is excellent tissue adhesiveness. This property simplifies the fixing of the gels[18,19]. More importantly, it allows conformal and intimate contact of the gel with the skin, which is essential for reliable signal transducing[4], whereas ordinary hydrogels are usually non-adhesive. Anti-freezing and anti-drying properties are indispensable for their long-term use in various environments. However, ordinary hydrogels will freeze at sub-zero temperatures, leading to loss of

[1]Key Laboratory of Functional Polymer Materials, Institute of Polymer Chemistry, College of Chemistry, Nankai University, Tianjin 300071, P. R. China. [2]School of Chemistry, Tiangong University, Tianjin 300387, P. R. China. ✉e-mail: yingguan@nankai.edu.cn; yongjunzhang@nankai.edu.cn

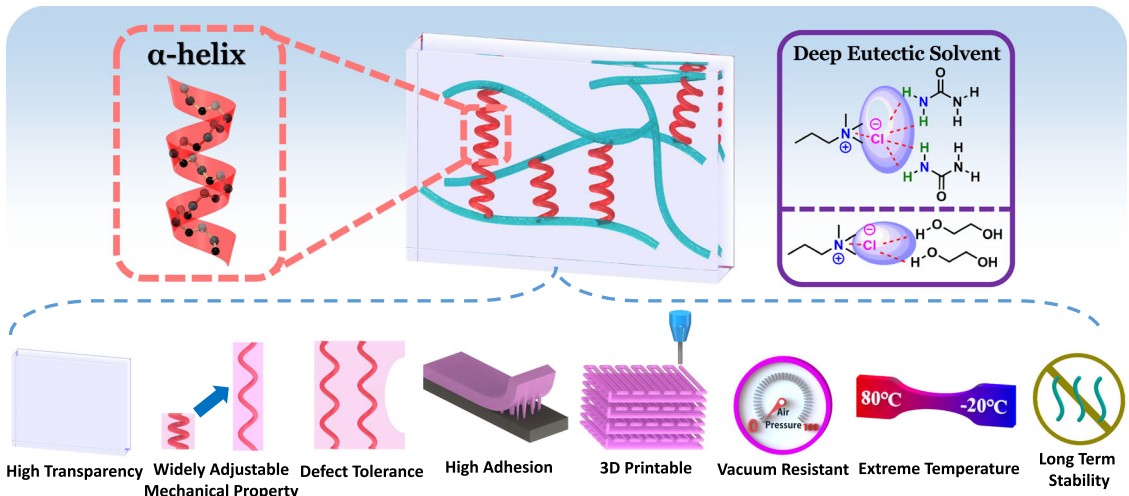

**Fig. 1 | Schematic diagram of the structure and properties of peptide-enhanced eutectogel.** The gel exhibits high toughness, high resilience, high adhesion, anti-drying, and anti-freezing properties simultaneously.

flexibility and conductivity[7,20]. They also lose water gradually, leading to changes in their mechanical and electronic properties. Significant efforts have been made to design conductive hydrogels with improved toughness[7], resilience[21], adhesion[17,22], anti-freezing[7,22,23] and anti-drying properties[21,24]. However, designing a gel with all these properties remains a huge challenge[22].

Here we report that conductive gels with all the properties required for high performance flexible strain/pressure sensors can be synthesized simply using a peptide cross-linker as cross-linker and deep eutectic solvent (DES) as solvent (Fig. 1). The peptide-crosslinked gels adopt a novel mechanism of energy dissipation through the breakage of intramolecular hydrogen bonds in the α-helical peptide chains. Like molecule-sized springs, these chains absorb energy when loading but return it back when unloading. Therefore the mechanical strength of the gels is improved significantly while maintains highly resilient simultaneously[25]. DESs are a new generation of green solvents sharing many characteristics and properties with ionic liquids but avoiding their major drawbacks, i.e., high toxicity, non-biodegradability, complex synthesis, and high cost[26]. Thanks to the good conductivity of DESs, the resulting eutectogels themselves are conductive without the need to add any other substances[27,28]. The low freezing point and low volatility of the solvent render the gels excellent anti-freezing and anti-drying properties. Interestingly, the gels also exhibit excellent adhesive property that are fully compatible with the adhesion to common surfaces. Combining all these properties, the gels give signals with unprecedented stability and repeatability when used as strain/pressure sensors.

## Results
### Synthesis of peptide crosslinkers and eutectogels
The eutectogels were synthesized by free radical polymerization of acrylamide (AAm) in the presence of a peptide cross-linker (PC) in DES (Fig. 2a). To synthesize the PCs (Supplementary Fig. 1), γ-benzyl-L-glutamate N-carboxyanhydride (BLG-NCA) was synthesized from BLG and triphosgene. Ring-opening polymerization of BLG-NCA was then carried out using 3-buten-1-amine as initiator. The amino end of the peptide was subsequently capped with acryloyl chloride, followed by deprotection of the benzyl groups in TFA. By adjusting the initiator to monomer ratio, three kinds of PCs were synthesized with polymerization degrees of 12, 22, and 32, namely PC12, PC22, and PC32, respectively. The chemical structures of the intermediates and final products were confirmed by [1]H NMR spectroscopy (Supplementary Figs. 2–5). The molecular weights of the products measured

by gel permeation chromatography (GPC) were collected in Supplementary Table 1.

To screen appropriate DESs, the solubility of PC22 was tested in a series of DESs using the most commonly used choline chloride (ChCl) as hydrogen bond acceptor (HBA) (Supplementary Table 2). It was found that the DESs with urea, ethylene glycol (EG), and glycerol as hydrogen bond donor (HBD) are good solvents for PC22, while PC22 dissolves poorly when ethanedioic acid, glutaric acid and glucose were used as HBDs. Heating improves the dissolving capacity of [ChCl][Urea], [ChCl][EG], [ChCl][Glycerol], and [ChCl][Glutaric Acid], while the dissolving capacity of [ChCl][Ethanedioic Acid] and [ChCl][Glucose] remains poor. The different dissolving capacity of the DESs may be ascribed to the different hydrogen bond donating capacity of the HBDs and the different viscosity of the DESs. Therefore, [ChCl][Urea] and [ChCl][EG] were selected as solvents for following studies. All 3 PCs dissolve well in the solvents, but the solubility decreases slightly with increasing molecular weight (Supplementary Fig. 6).

Similar to the synthesis of hydrogels, the eutectogels were synthesized by dissolving the monomer AAm, the cross-linker PC, and the photo-initiator DEAP in DESs and then exposed to UV light[29]. Adding AAm actually forms a ternary DES as evidenced by further decrease in freezing point. (Supplementary Fig. 7) The resulting gels were named as $A_xB_y$, where A represents the monomer AAm, x represents the mass ratio of monomer to solvent, B represents the cross-linker, y represents the molar ratio of cross-linker to monomer. For example, $A_{0.4}PC22_{1\%}$ means the mass ratio of AAm to DES is 0.4, the cross-linker is PC22, and the molar ratio of PC22 to AAm is 1%. The AAm contents remaining in the gels was measured by HPLC to be 0.0359 wt%. The conversion rate of the AAm monomer was thus determined to be 99.86%. FTIR and [13]C solid-state NMR studies revealed a similar chemical structure of peptide-crosslinked gels with the BIS-crosslinked ones (Supplementary Figs. 8 and 9). The amount of unreacted double bonds left in the gels was negligible. The peptide-crosslinked gels also present thermal behaviors similar to that of BIS-crosslinked gels, as demonstrated by DSC and TGA studies. (Supplementary Figs. 10 and 11) The moisture content in the eutectogel was determined to be 2.77% by Karl-Fischer coulometric titrations[30].

The resulting eutectogels are highly transparent (Fig. 2b). The transmittance at 600 nm reaches up to 94.0% for the gel in [ChCl][EG] and 92.3% for the gel in [ChCl][Urea] (gel thickness: 2 mm) (Fig. 2c). The result suggests that like the PC and monomer AAm, the resulting PAAm chains also have a high miscibility with the DESs. In addition to the conventional mold forming method, the gel can also be formed by

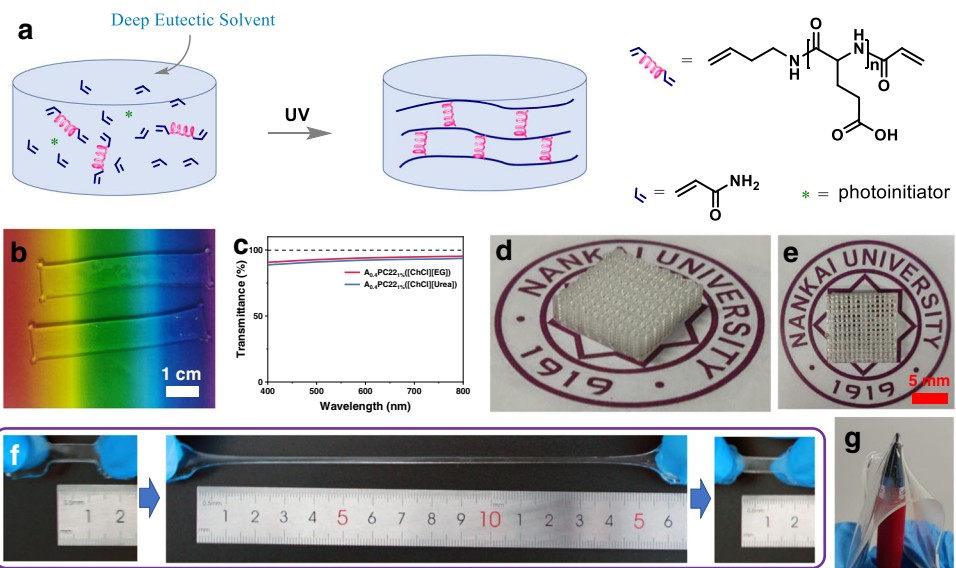

**Fig. 2 | Synthesis and characterizations of peptide-enhanced eutectogel.**
**a** Dissolving monomer, PC and photo-initiator in DES followed by photo-polymerization. **b** Photograph and **c** transmission spectra of the two eutectogels. **d**, **e** 3D-printed $A_{0.4}PC22_{1\%}$ [ChCl][EG] gel. **f** Stretching a gel to 1500% strain without failure. **g** Pushing a gel film against a tip.

3D printing (Figs. 2d and 2e). As 3D printing ink, the viscosity of the pre-gel solutions was determined to be ~175 mPa s, making them suitable ink for 3D printing[31]. They are also stable when stored at 4 °C in a refrigerator (Supplementary Figs. 12 and 13). The resolution of the resulting 3D structure is 410 μm in x direction and 100 μm in y direction. More importantly, the eutectogels exhibit extraordinarily strong mechanical properties. As a demonstration, Fig. 2f shows that an eutectogel could be stretched to 1500% strain without failure. No damage was observed after pressing a gel film firmly against a sharp tip (0.5 mm in diameter) (Fig. 2g and Supplementary Movie 1).

## Mechanical properties of the eutectogels

Tensile tests were carried out to further study the mechanical properties of the gels using [ChCl][EG] as solvent. Unlike peptide-crosslinked eutectogels, the eutectogel crosslinked with the conventional cross-linker BIS is mechanically weak with an elongation at break of 260% and a tensile strength of 0.14 MPa (Fig. 3a). When it is replaced by equimolar PC, both elongation at break and tensile strength increase dramatically. Particularly for the PC22-crosslinked eutectogel, the elongation at break increases to 1860%, corresponding to over 6 folds increase compared with that of the BIS-crosslinked gel. Meanwhile, the tensile strength increases to 1.04 MPa, which is over 6 folds higher than that of the BIS-crosslinked gel. Interestingly the elastic modulus (stiffness) also increases substantially when the gels are crosslinked with PC12 and PC22 (Fig. 3b). More importantly, cross-linking with PCs significantly improve the toughness of the gels. For example, the toughness of the PC22-crosslinked eutectogel was calculated to be 10.6 MJ m$^{-3}$ from the area under the stress-strain curve, corresponding to ~45 folds increase compared with that of the BIS-crosslinked gel (0.232 MJ m$^{-3}$).

Like the PC-crosslinked hydrogels, the significantly improved mechanical properties of the PC-crosslinked eutectogels may be attributed to a reduced inhomogeneity of the networks[25]. The inhomogeneous network of ordinary hydrogels is considered as a leading reason for their brittleness, because an inhomogeneous network cannot respond to external tensions in a cooperative way and thus leads to fracture at a low strain[6,32]. Compared with the BIS-crosslinked gel, the PC-crosslinked gel will have a lager mesh size because of the incorporation of the long peptide chain into the network. Therefore the network will be less inhomogeneous and can resist a larger strain.

The hypothesis is confirmed by the fact that the gel becomes more ductile when it is crosslinked with a longer PC (Fig. 3a). The incorporation of peptide chains into the gel network not only reduces the inhomogeneity of the networks, but also provides a novel mechanism for energy dissipation[25]. The lack of energy dissipation mechanism is well known to be another major reason for the poor mechanical properties of conventional hydrogels[6,11]. As revealed by CD spectra (Supplementary Fig. 14) the peptide cross-linker dissolved in the DESs adopts an α-helical structure, which is evidenced by the two minima at 213 and 225 nm. More importantly the peptide chains incorporated in the eutectogels also adopt an α-helical structure. (Supplementary Figs. 14 and 15) It is well-known that the α-helical structure is stabilized by hydrogen bonds between the C = O group of each amino acid and the NH group of amino acid four residues earlier in the sequence[33]. These intramolecular hydrogen bonds can be fractured by external force, and thus dissipating energy applied on the gel[34]. According to previous AFM study, the energy required to break the intramolecular hydrogen bonds is 20.2 kJ/mol[34]. The breakage of the intramolecular hydrogen bonds leads to the destruction of the α-helical structure, which was evidenced by significant change in the CD spectra of the streched gel (Supplementary Fig. 15).

Besides the length of the PC, its content also influences the mechanical property of the gel. When the PC22 content increases from 0.25% to 1.5%, the elongation at break gradually decreases from 2400% to 750%, while the elastic modulus increases from 0.0173 to 0.371 MPa (Fig. 3c, d). However, the tensile strength and toughness first increase with increasing PC22 content but then decrease with further increasing PC22 content. Like hydrogels, increasing cross-link density improves the gel strength but reduce its extendibility[25,35]. The AAm content also has influence on the mechanical properties of the gel. Increasing AAm content in the gel leads to increasing elastic modulus but reduced elongation at break, revealing that the gel becomes stiffer but less ductile (Fig. 3e, f). As a result, the tensile strength and toughness of the gel first increase with AAm content but then drop when further increasing AAm content.

The eutectogels using [ChCl][Urea] as solvent behave similarly to those using [ChCl][EG] as solvent (Supplementary Fig. 16). Replacing BIS with PC dramatically improves the mechanical properties of the gel, as indicated by the significantly increased elongation at break, tensile strength and toughness (Supplementary Figs. 16a and 16b). The

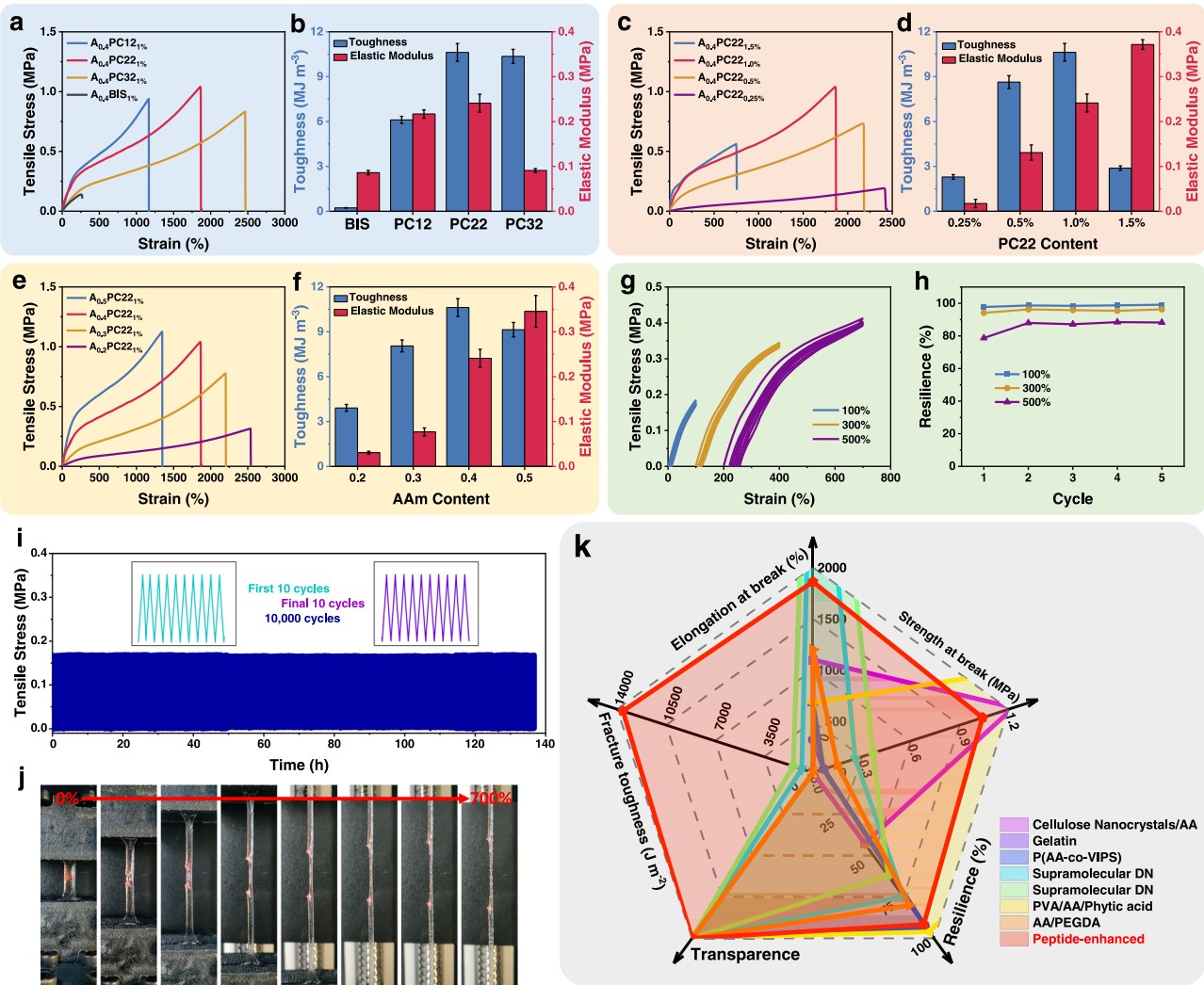

**Fig. 3 | Tensile mechanical properties of the eutectogels with [ChCl][EG] as solvent. a, b** Tensile stress-strain curves (**a**) and mechanical properties (**b**) of gels crosslinked with different crosslinkers. **c, d** Tensile stress-strain curves (**c**) and mechanical properties (**d**) of gels crosslinked with various amount of PC22. **e, f** Tensile stress-strain curves (**e**) and mechanical properties (**f**) of the PC22-crosslinked gels with various contents of AAm. **g** Stress-strain curves of $A_{0.4}PC22_{1\%}$ gel during 5 consecutive loading-unloading cycles at a maximum strain of 100%, 300%, and 500%. The curves at a maximum strain of 300% and 500% were shifted along the x axis for clarity. **h** Resilience of the gel. **i** Stress of the gel during 10,000 consecutive loading-unloading cycles at a maximum strain of 100%. The inset is first 10 cycles and final 10 cycles. **j** Photos of a $A_{0.4}PC22_{1\%}$ gel with a 2.5 mm crack at different tensile strains. The crack was stained with rhodamine B for visibility. The width of the gel sample was 5.0 mm. **k** Mechanical properties of the peptide-enhanced eutectogel compared with other eutectogels reported in the literatures. The error bars indicate standard deviation.

effects of PC length, PC content, and AAm content on the mechanical properties of the gel are also similar to the gel using [ChCl][EG] as solvent (Supplementary Fig. 16c–f). However, the elastic modulus, tensile strength, and toughness are slightly lower than the gel using [ChCl][EG] as solvent. Among all the eutectogels studied here, the $A_{0.4}PC22_{1\%}$ gel with [ChCl][EG] as solvent presents the best overall mechanical properties with a large elongation at break (>1800%), a high tensile strength (>1.0 MPa) and a high toughness (>10 MJ m$^{-3}$) and therefore was chosen for the following studies.

The peptide-enhanced eutectogels are not only extremely tough and greatly stretchable, but also highly resilient, which is critical for their applications as strain sensors[36]. As an example, cyclic tensile tests were performed on $A_{0.4}PC22_{1\%}$ gel with [ChCl][EG] as solvent (Fig. 3g). A very small hysteresis loop was observed in each cycle, indicating only a small amount of energy was dissipated as heat. At a maximum strain of 100%, only 2.25% of the energy was dissipated as heat in the first cycle. This value was reduced to 1.34% in the second cycle and further decreased to 0.88% in the fifth cycle. Similar results were observed

when testing at a maximum strain of 300% and 500%, although the ratio of energy dissipated as heat increased slightly compared with that at a maximum strain of 100%.

Since the gel has a low energy loss when deformed, it exhibits a high resilience. As shown in Fig. 3h, the resilience of the gel was calculated to be 97.8%, 94.1%, and 78.7% in the first loading-unloading cycle at a strain of 100%, 300%, and 500%, respectively. This value increases to be 98.7%, 96.3%, and 88.0% in the second cycle and finally approaches to 99.1%, 96.3%, and 88.3%. This value is close to that of the well-known highly resilient materials, including resilin in dragonfly tendon (92–97%), elastin in human skin and arteries (~90%), and polybutadiene rubber (80%)[36].

The high resilience of the gel is attributed to its unique mechanism for energy dissipation[25]. Previously to improve the mechanical properties of hydrogels, numerous mechanisms for energy dissipation via the fraction of covalent or dynamic bonds was introduced. However, these bonds cannot reform or only reform partially as tensile unloading, leading to a large fraction of energy is dissipated as heat

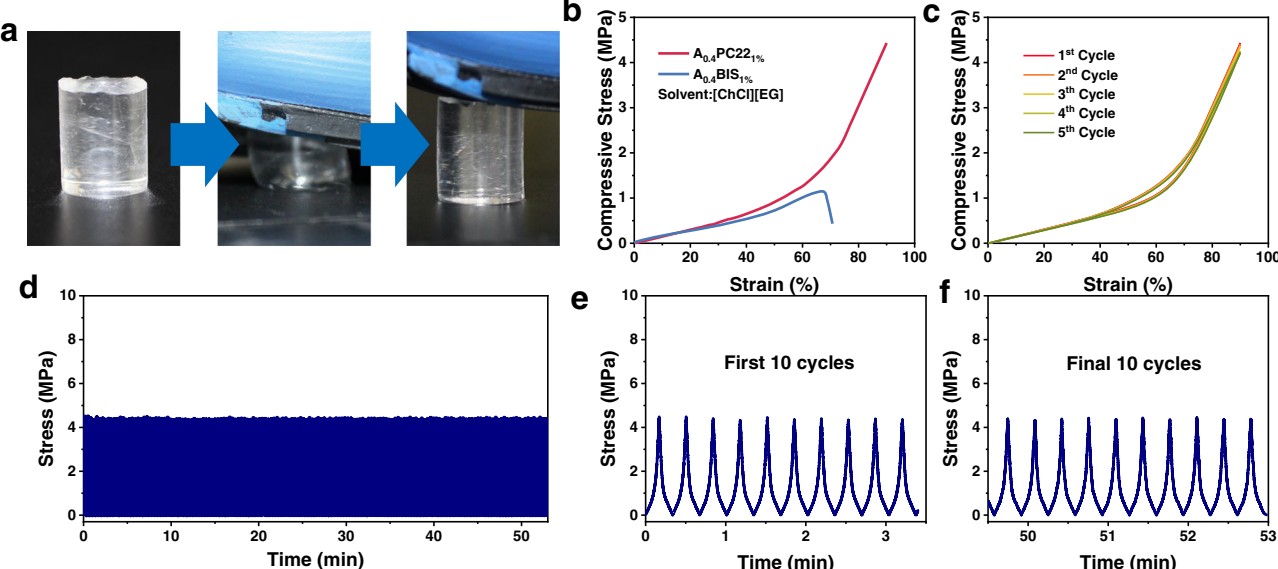

**Fig. 4 | Compression mechanical properties of the eutectogels with [ChCl][EG] as solvent. a** No damage occurred after stepping strengthly on a gel. **b** Compressive stress-strain curves of gels crosslinked with PC22 and BIS. **c** Stress-strain curves of $A_{0.4}PC22_{1\%}$ gel during 5 consecutive compression. **d** Stress of the gel during 150 consecutive compressive loading-unloading cycles. **e** The first 10 cycles. **f** The final 10 cycles.

and a large hysteresis loop on their stress-stain curves[11,13,14,37]. Recently some tough eutectogels were developed by adopting similar energy dissipation mechanisms. Like their hydrogel analogs, a large hysteresis loop was observed on their stress-stain curves[38,39]. Very differently for the peptide-enhanced eutectogels developed here, when unloaded, the peptide chains will fold back from an extended structure to helical structure. As the peptide chains fold back precisely, the fractured intramolecular hydrogen bonds reform almost completely, thus returns the energy absorbed during the loading process to the system. As a result, only a very small hysteresis loop is observed. The recovery of CD spectra when the gel is unloaded confirms the reformation of the helical structure (Supplementary Fig. 15).

The high fatigue resistance of the gel is also reflected by the fact that the stress-strain curves almost overlap each other when tested at all three maximum strains (Fig. 3g). It is noteworthy that the loading-unloading cycles were repeated without waiting. In contrast, for many of the previously reported gels, the stress-strain curve cannot fully recover even after a long waiting time[11,13,14,37]. To further demonstrate the extraordinary fatigue resistance of the gel, it was subjected to 10,000 consecutive stretching/releasing cycles at 100% strain (Fig. 3i). It is clear that no stress weakening occurs and that the final 10 cycles are almost identical to the initial 10 cycles. The extraordinary environmental stability should be attributed to the anti-drying properties of the gel (see below). In addition, the stretching responses of the eutectrogel using both solvents are almost independent of the stretching rate (Supplementary Fig. 17)[13,37].

The eutectogels are also highly crack-insensitive. As an example, a 2.5 mm crack was made on an $A_{0.4}PC22_{1\%}$ [ChCl][EG] gel sample with a width of 5.0 mm and stained with rhodamine B for visibility (Fig. 3j and Supplementary Movie 2). At the initial stage of stretching, the two sides of the crack were stuck together because of the self-adhesive properties of the gel (see below). They did not separate until a ~240% strain was reached. The crack did not propagate with stretching. As a result, an elongation at break of 730% was observed for the sample. In contrast, when stretching a BIS-crosslinked eutectogel with a 0.5 mm crack, the crack propagated quickly. The elongation at break was less than 50% (Supplementary Fig. 18 and Supplementary Movie 3). From the above tests, a fracture toughness as high as 13,600 J m$^{-2}$ was calculated for $A_{0.4}PC22_{1\%}$ [ChCl][EG] gel by the compliance method of

Greensmith et al.[40]. To the best of our knowledge, this is the highest value for single network gels and is even comparable to that of double network gels (Supplementary Table 3). The fracture toughness of the $A_{0.4}BIS_{1\%}$ [ChCl][EG] gel was also determined, which is only 38 J m$^{-2}$. One can see that replacing BIS with PC22 dramatically increases the fracture toughness of the gel by 3 orders of magnitude. For comparison, the mechanical properties of eutectogels previously reported in the literatures[39,41–46] are summarized in Fig. 3k and Supplementary Table 4.

Replacing BIS with PC also significantly improves the compressive strength of the eutectogels. After being trampled with foot, the gel could restore its original shape completely and immediately (Fig. 4a). Figure 4b compares the behaviors of $A_{0.4}PC22_{1\%}$ and $A_{0.4}BIS_{1\%}$ gels both using [ChCl][EG] as solvent. The peptide-crosslinked gel remained undamaged even when pressed to the largest compressive strain of the machine (90%) at a compressive stress of 4.5 MPa. As a comparison, the gel crosslinked with BIS was impaired at a compressive strain of 67% and a compressive stress of 1.16 MPa. High resilience was again observed when the peptide-crosslinked gel was subjected to five consecutive cycles of compression tests at a maximum compressive strain of 90% (Fig. 4c). In each cycle, the loading-unloading curve almost overlapped with the previous one, which is different from gel materials with other energy dissipation mechanisms[35]. These results suggest no permanent damage to the gel network during the compression process and the compressed gels could return to its initial state within the time scale of the experiment. The compression fatigue resistance of the peptide-enhanced eutectogel was further demonstrated by 150 consecutive compression cycles at 90% strain (Fig. 4d). In each cycle, the maximum compressive stress at 90% strain keeps almost constant at ~4.4 MPa, revealing robust fatigue resistance of the gel (Fig. 4d–f).

## Adhesion properties of the eutectogels

It is interesting that the peptide-enhanced eutectogels with [ChCl][EG] as solvent also present excellent self-adhesive properties. The $A_{0.4}PC22_{1\%}$ [ChCl][EG] gel can adhere tightly to various items, including weight, Buchner funnel, suction bulb, box, and beaker, and lift them up, demonstrating excellent adhesion on a broad range of substrates including metal, ceramics, rubber, paper, and glass (Fig. 5a).

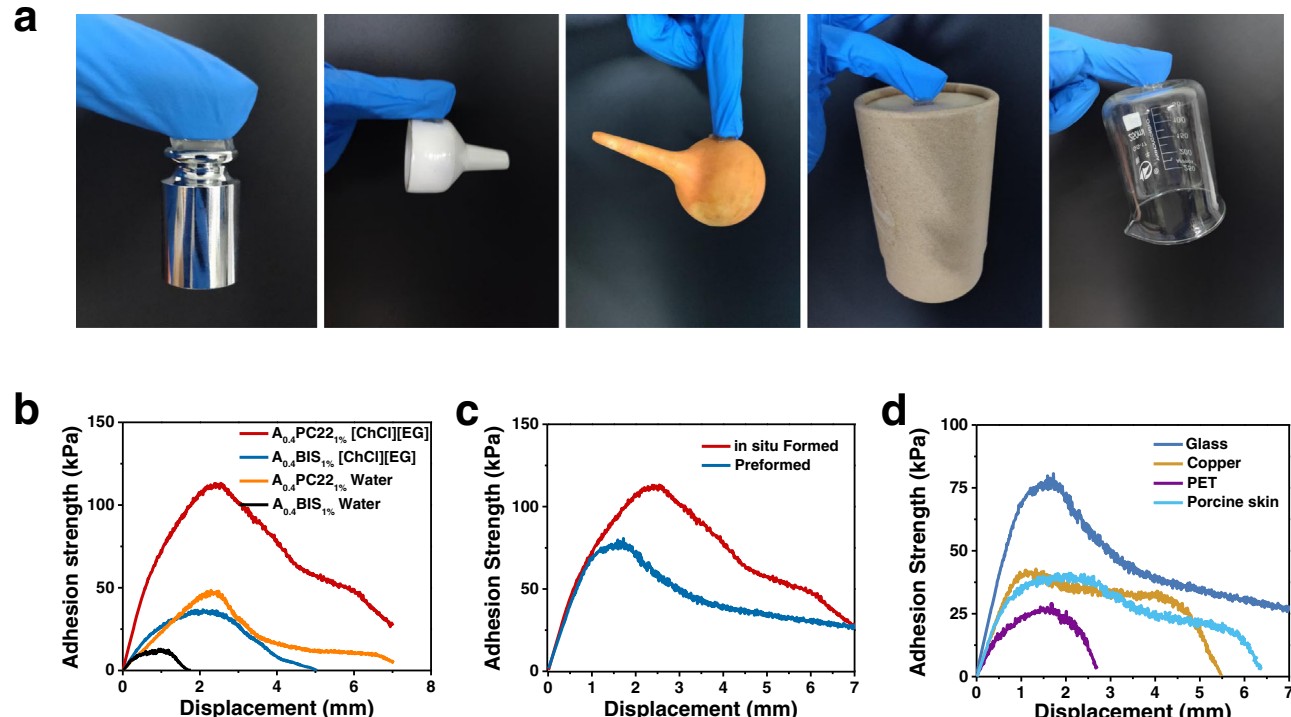

**Fig. 5 | Adhesive properties of gels. a** Sticking and lifting of a weight (100 g), a Buchner funnel (~60 g), a suction bulb (~32 g), a box (~102 g), and a beaker (~109 g) with $A_{0.4}PC22_{1\%}$ [ChCl][EG] gel. The mass of the gels was ~0.27 g. **b** Sheer peeling curves of in situ formed eutectogels and hydrogels. **c** Sheer peeling curves of in situ formed and preformed $A_{0.4}PC22_{1\%}$ [ChCl][EG] gels. **d** Sheer peeling curves of preformed $A_{0.4}PC22_{1\%}$ [ChCl][EG] gels on various substrates.

The adhesion strength of the gels was measured by lap shear test using glass slides as adherends[47]. As shown in Fig. 5b, the adhesive property of $A_{0.4}BIS_{1\%}$ hydrogel is relatively poor with an adhesion strength of ~12.8 kPa. Replacing the solvent from water to [ChCl][EG] significantly improved the adhesive property of the gel (adhesion strength: ~35.5 kPa). Recently Liang et al.[39] also reported a double network (DN) eutectogel can stick to different substrates with an adhesion strength of 13.92 kPa on glass. Replacing BIS with PC22 also leads to improved adhesive property. Compared with $A_{0.4}BIS_{1\%}$ hydrogel, the adhesion strength of $A_{0.4}PC22_{1\%}$ hydrogel increases by about 3 folds, reaching ~48.4 kPa. The improved adhesive property of the peptide-crosslinked hydrogel can be attributed to the introduction of a large amount of adhesive groups, such as carboxyl groups on the PC[48]. The adhesive property of the $A_{0.4}PC22_{1\%}$ [ChCl][EG] gel is further improved (~112.4 kPa) compared with $A_{0.4}BIS_{1\%}$ hydrogel, because both the solvent is changed from water to a DES, and the cross-linker is changed from BIS to PC22.

In the above measurements the gel samples were synthesized in situ between glass slides. The adhesive property of preformed gel, which is more commonly used in practical applications, was also measured by lap shear test. As shown in Fig. 5c, the adhesion strength of the preformed gel is lower than the one formed in situ, but is still as high as ~77.6 kPa. Unlike the in situ formed gel, it is difficult for the adhesive groups in the preformed gel to move to the interface to form bonds with the substrate, leading to a reduced adhesion[14,49]. Even so, the preformed gel is still highly adhesive to many common substrates, with an adhesion strength of ~41.5 kPa on copper, ~39.7 kPa on porcine skin, and ~27.0 kPa on PET (Fig. 5d).

## Performance under extreme conditions

A big problem for hydrogel-based flexible devices is that they will lose their elasticity at sub-zero temperatures because of the freezing of water, which severely limits their applications in cold conditions. In contrast, because of the low freezing point of [ChCl][EG], the gel using it

as solvent reveals excellent anti-freezing properties. As demonstrated in Supplementary Movie 4 and Fig. 6a, after being immersed an ice-salt bath (~ −27 °C) for 30 min, the gel still maintains soft and can be twisted. Figure 6b compares the tensile properties of the gel at various temperatures. At 20 °C, the gel can be stretched to a strain of 1860%. When cooled to −20 °C, the value is still as high as 1590%. Meanwhile, the tensile strength only slightly increases from 1.04 MPa to 1.16 MPa when cooled from 20 °C to −20 °C. Generally, with increasing temperature, the elongation at break increases while the tensile strength decreases, which may be attributed to the decreased viscosity of the solvent[50] and increased mobility of the polymer chain at high temperature. DSC study further confirms that the freezing point of the gel is −51.2 °C, lower than the freezing point of pure [ChCl][EG] (−36.3 °C)[51] (Fig. 6c). In addition, no glass transition or phase transition was found in the temperature range from 80 °C to −50 °C. Therefore, little change in mechanical properties of the gel was found in the wide temperature range.

Another severe problem for hydrogel-based flexible devices is their instability due to their gradual dehydration. Like ionic liquids, DESs have an ultra-low partial vapor pressure, so that eutectogels are expected to exhibit a long-term stability[26]. As shown in Fig. 6d, after 30 days storage under ambient conditions, a mass loss of only 1.4% was found for the eutectogel with [ChCl][EG] as solvent. Very differently the hydrogel sample lost weight quickly, and its mass loss reached 55% after 7 days storage. In addition, the eutectogel kept highly transparent after 30 days storage under ambient conditions (Fig. 6e). Visual examination reveals no detectable change in size and appearance for the two eutectogels after 30 days storage, while the hydrogel sample shrank dramatically and became no longer transparent (Fig. 6f). More importantly little change in tensile properties was found for the eutectogel. After 30 days storage under ambient conditions, the elongation at break and tensile strength of the gel decreased by only 5.1% and 4.9%, respectively (Fig. 6g). The high anti-drying properties of the gel was further confirmed by 3 days storage under medium vacuum (~40 Pa). The gel remained to be flexible (Supplementary

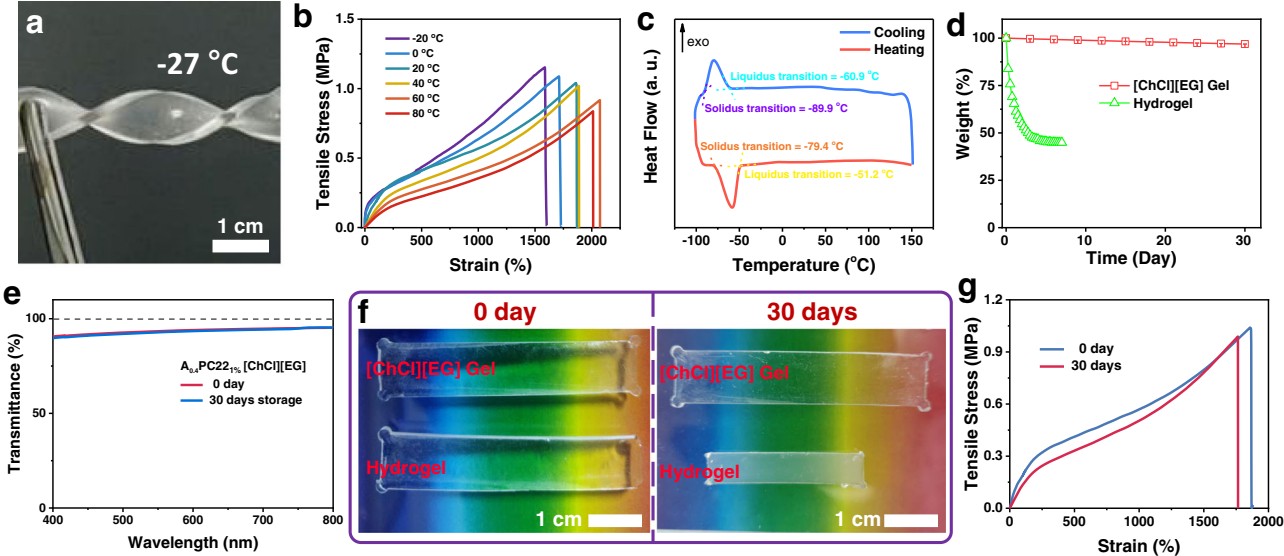

**Fig. 6 | Anti-freezing and anti-drying properties. a** Twisting an $A_{0.4}PC22_{1\%}$ [ChCl] [EG] gel after cooled to −27 °C. **b** Tensile stress-strain curves of an $A_{0.4}PC22_{1\%}$ [ChCl] [EG] gel at various temperatures. **c** DSC curve of an $A_{0.4}PC22_{1\%}$ [ChCl][EG] gel. **d** Weighs of an $A_{0.4}PC22_{1\%}$ [ChCl][EG] gel and a hydrogel stored under ambient conditions. **e** Transmission spectra of an $A_{0.4}PC22_{1\%}$ [ChCl][EG] gel before and after 30 days storage under ambient conditions. **f** Photos of an $A_{0.4}PC22_{1\%}$ [ChCl][EG] gel and a hydrogel before and after 30 days storage under ambient conditions. **g** Tensile stress-strain curves of an $A_{0.4}PC22_{1\%}$ [ChCl][EG] gel before and after 30 days storage under ambient conditions.

Fig. 19) and little change in its appearance, transparency, and tensile properties were found (Supplementary Figs. 20–22).

## Application as strain/pressure sensors

Unlike water, DESs themselves are ionically conductive, therefore eutectogels are conductive themselves without the need for adding salts. The conductivity of $A_{0.4}PC22_{1\%}$ [ChCl][EG] gel was measured to be 3.82 mS cm$^{-1}$ at room temperature, making it possible to be used as a resistivity-type strain/pressure sensor.

Its potential to be used as tensile strain sensor was first tested. Stretching the gel narrows the channel for ionic migration and meanwhile elongates the migration path, leading to increased resistance of the gel with increasing tensile strain (Supplementary Fig. 23a). The sensor exhibits a relatively high sensitivity, as indicated by a gauge factor (GF) of 1.74 at 0–200% strain, 2.35 at 200–400% strain, and 3.17 at 400–1000% strain. Upon tensile loading, the sensor responds quickly with a response time of ~165 ms. It recovers quickly upon unloading with a recovery time of ~155 ms (Supplementary Fig. 23b). More importantly, the sensor gives highly stable and reliable resistance signals. When repeatedly stretched to a same strain, almost the same resistance signals were given (Supplementary Fig. 23c). In addition, the resistance signal is independent of the stretching speed. Almost the same resistance signal was produced when the gel was stretched at different stretching speeds under same strain (Supplementary Fig. 23d).

The gel can also be used to detect compressive stress (pressure). The current of the gel rises with increasing compression strain, because the ion migration path is shortened and the channels for ion migration are widened (Supplementary Fig. 24a). Similar to hydrogel-based pressure sensors, the current decreases sharply in the low-pressure range, indicating a high sensitivity of the sensor in this pressure range. The response of the sensor is quite fast, with a response time of ~350 ms upon loading and a recovery time of ~400 ms upon unloading (Supplementary Fig. 24b). The current signals were reproducible (Supplementary Fig. 24c) and independent of the pressing rate (Supplementary Fig. 24d).

The potential of the gel as a wearable device to monitor human motion was then tested. Thanks to its good adhesiveness, the gel can be facilely attached to human body by itself without the help of a tape or bandage. Such intimate and conformal contact between the gel and skin is highly desirable as it will allow precise detection. A piece of gel was attached on a finger to monitor its motion (Fig. 7a). Bending of the finger stretches the gel, leading to an increased resistance. In addition, the $\Delta R/R_0$ value increases with increasing bending angle of the finger. When bending the finger to a same angle at different bending speeds, almost the same electronic signals were detected, indicating high reliability of the sensor (Fig. 7b). By attaching the gel to wrist and knee, their movements can be accurately monitored in the same way (Fig. 7c and d). The high sensitivity of the sensor allows it to monitor some subtle muscle movements, such as speaking. For this purpose, a gel was attached to the volunteer's throat (Fig. 7e). Different signal patterns were detected when the tester spoke different words, however, when the tester spoke the same word, a similar pattern was always detected.

To further demonstrate the excellent performance of the eutectogel-based sensor, a robotic finger was employed, which allows precise control of bending level, long-term tests and tests under extreme conditions (Fig. 8a). The PWM signal value of the servo motor is used to represent the position of the robotic finger, and the relevant level of bending can be visualized in Supplementary Fig. 25. As shown in Fig. 8b, the bending of the finger stretches the gel and therefore increases the resistance. Higher bending level gives a larger resistance signal. On the other hand, the bending level of the finger can be revealed from the resistance signal. Almost the same resistance signal is given when the finger is bent to the same level, indicating high stability of the sensor. The signal stability was also confirmed by the fact that it is independent of the bending speed (Fig. 8c). To further evaluate the long-term stability and durability of the sensor, 10,000 consecutive bending cycles were performed under ambient conditions (Fig. 8d). The whole test lasted over 106,000 s, i.e., 29.5 h. One can see the resistance signals were highly reproducible in each cycle. As examples, the results of the first 10 cycles and the final 10 cycles were presented in the inset of Fig. 8d. One can see both the peak values and the curve shapes maintain the same even after 10,000 bending cycles. It is noteworthy the long-term

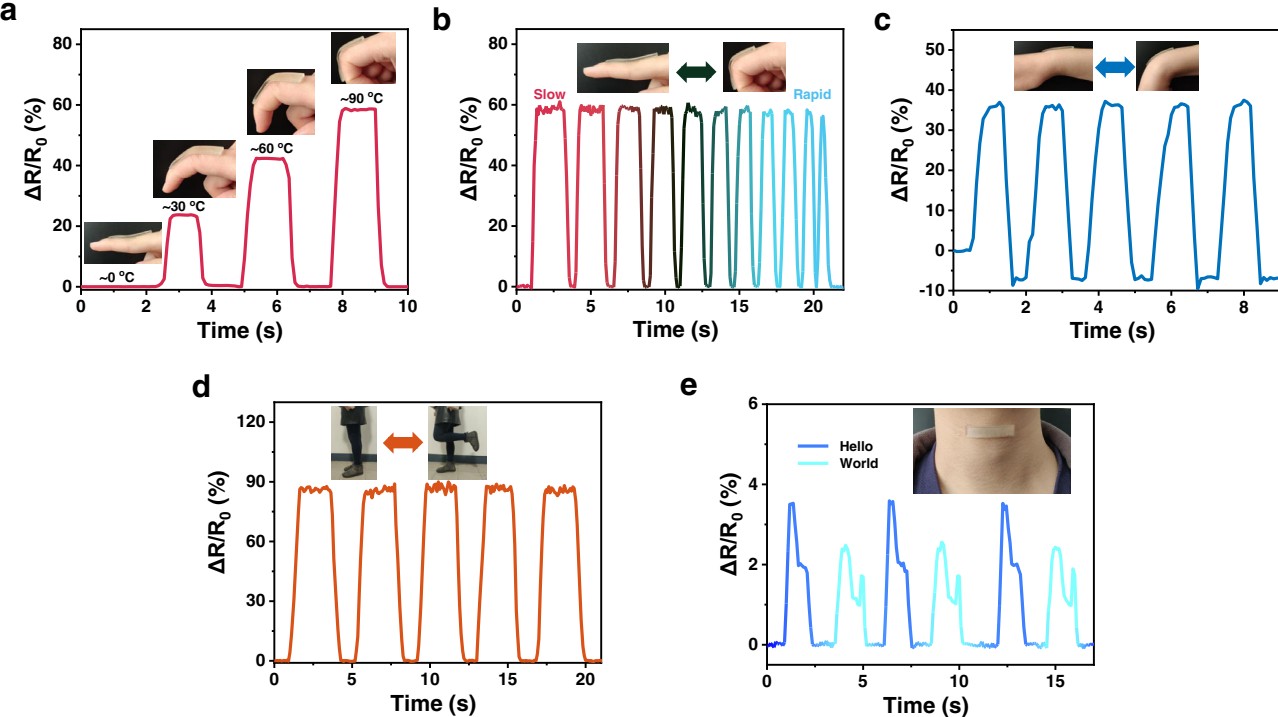

**Fig. 7 | Human motion monitoring. a, b** Monitoring finger bending to various angle (**a**) and different speeds (**b**). **c–e** Monitoring wrist bending (**c**), knee movement (**d**), and speaking (**e**). An $A_{0.4}PC22_{1\%}$ [ChCl][EG] gel was attached to the appropriate site to monitor the movements.

signal stability achieved here is unprecedented. For many previously reported gel-based strain sensors[52–54], including the recently-reported eutectogel-based ones[39], obvious variation of the signals and baseline shifts were usually observed. The extraordinary stability of the sensor is mainly attributed to the high resilience of the gel and the low vapor pressure of the solvent. Furthermore, the excellent anti-freezing and solvent-retention properties of the eutectogel allow the sensor not only work at room temperature, but also at low and high temperatures (Fig. 8e). Highly stable resistance signals were again obtained when repeatedly measured at both −20 °C and 80 °C. Only a very slight decrease in the peak value was observed when temperature drops from 80 °C to −20 °C, which is consistent with the slightly increased modulus of the gel as shown in Fig. 6b.

The performance of the eutectogel as a pressure sensor was tested by pressing the gel with a robotic finger (Fig. 8f). Pressing the gel with the robotic finger increases the current of the gel (Fig. 8g). The current signals maintain almost identical at the same pressure, indicating high reliability of the sensor (Fig. 8h). The highly stable signal was further proofed by 10,000 consecutive press tests (Fig. 8i). The signal in each cycle maintains the same. No baseline drift was observed, which is quite different from the previously reported gel-based sensors[54]. The pressure sensor also works well at both low temperature (−20 °C) and high temperature (80 °C) with highly reproducible resistance signals (Fig. 8j). Like the strain sensor, a very small decrease of the signal was observed because of the slightly increased modulus of the gel when cooled.

To further demonstrate the practical application potential of the eutectogels, we designed an obstacle avoidance scenario. As shown in Supplementary Movie 5 and Supplementary Fig. 126a–c, when the moving robotic finger installed with the gel sensor touches a human finger, the robotic finger will sense the obstacle and quickly lift up to avoid it, thus avoiding further collision. In contrast, robotic finger without the sensor will continue to move and push the human finger away, which may result in safety problems. The implementation of this function relies on the resistance change of the gel sensor when it touches the obstacle. The specific logic architecture was shown in Supplementary Fig. 26d, where the sensor was installed under the robotic finger and its signal was transmitted directly to the analog port of an Arduino controller after passing through a voltage divider circuit. The Arduino then controls the lifting of the robotic finger when it determines that the voltage has increased. In this experiment, the voltage change threshold was set to be 10%. Therefore, this sensor has great application prospects in robot remote monitoring and intelligent electronic skin.

## Discussion

In summary, a conductive gel combining properties including excellent stretchability, high toughness, high resilience, excellent adhesive property, anti-freezing and anti-drying was successfully synthesized by photopolymerization of AAm in the presence of PCs in DES. Replacing of the common cross-linker with PC dramatically enhances the mechanical properties of the gel. Thanks for its unique mechanism for energy dissipation, the gel remains highly resilient. The gel is also highly crack-resistant, with the highest fracture toughness reported up to now for single network gels. DSC study revealed that the DES has a low freezing point. No glass transition or phase transition occurs for the gel in the temperature range from 80 °C to −50 °C. As a result little change in mechanical properties of the gel was found in the wide temperature range. Particularly the gel still remains flexible at −27 °C. The low volatility of DES renders the gel excellent solvent-retention properties. Only a very small mass loss was found for the eutectogel after 30 days storage under ambient conditions or 3 days under medium vacuum. The gel is also highly transparent and adhesive. In addition, it is conductive due to the high conductivity of DES. All these properties make the eutectogel excellent for wearable, flexible strain/pressure sensor. The sensor

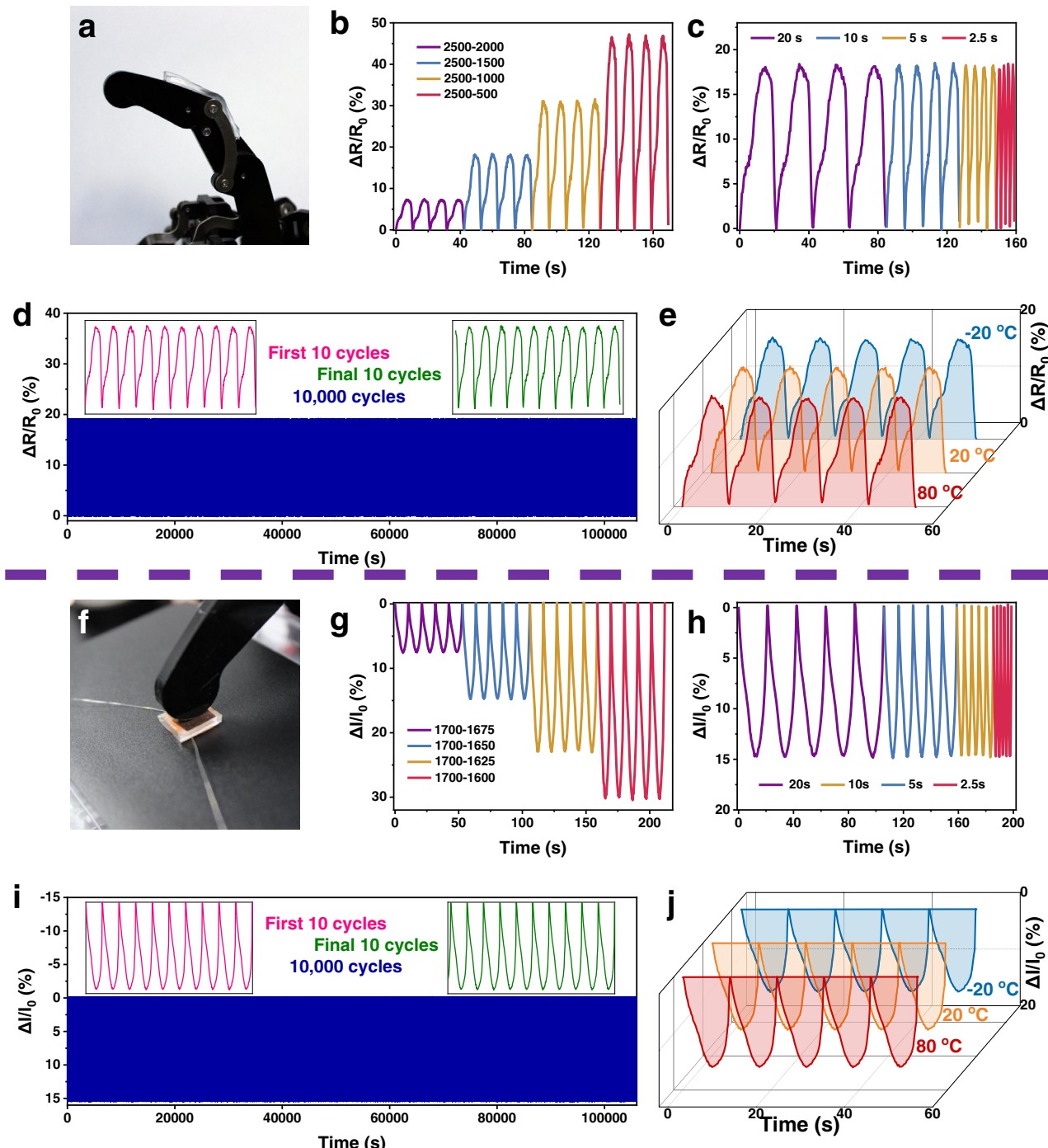

**Fig. 8 | Tests with robotic finger. a** A gel was attached on the robotic finger to monitor its bending. **b** Relative resistance changes under different bending levels. **c** Relative resistance change at different bending speeds (bending level: 2500-1500). **d** Relative resistance changes during 10,000 consecutive finger bending with the bending level is 2500-1500. The insets are the first 10 cycles and the final 10 cycles. **e** Consecutive finger bending at various temperatures. Bending level: 2500-1500. **f** Pressing a gel sample with robotic finger. **g** Relative resistance changes under different pressing levels. **h** Relative resistance changes at different pressing speeds (pressing degree: 1700-1650). **i** Relative resistance changes during 10,000 consecutive finger pressing with the pressing degree is 1700-1650. The insets are the first 10 cycles and the final 10 cycles. **j** Consecutive pressing at various temperatures. Pressing degree: 1700-1650.

works well over a wide range of temperatures (from −20 °C to 80 °C). More importantly, it is highly reliable and durable. As a demonstration, in a test composed of 10,000 consecutive bending cycles which lasted for ~30 h under ambient conditions, the sensor gave almost the same resistance signals in each cycle. Finally, an intelligent obstacle avoidance function demonstrates potential applications including remote monitoring of robots and intelligent electronic skin.

## Methods

### Materials

γ-Benzyl-L-glutamate (BLG), triphosgene, 4-dimethylaminopyridine (DMAP), trichloroacetic acid (TFA), *N,N*-dimethyl formamide (DMF) and tetrahydrofuran (THF) were purchased from Tianjin Heowns Biochem LLC. 1-Hydroxybenzotriazole (HOBT), 3-buten-1-amine, 2,2-diethoxyacetophenone (DEAP) and hydrobromic acid aqueous (48% HBr) were purchased from Sigma-Aldrich. Acryloyl chloride,

acrylamide (AAm) and *N,N*-methylenebisacrylamide (BIS) were purchased from Innochem Co. Choline chloride (ChCl), ethylene glycol (EG), urea, glycerol, ethanedioic acid, glutaric acid and glucose were purchased from Aladdin.

### Synthesis of γ-benzyl-L-glutamate N-carboxyanhydride (BLG-NCA)

BLG (5.0 g, 21 mmol) and triphosgene (4.0 g, 14 mmol) were dissolved in dried THF (50 mL) with magnetic stirring in nitrogen atmosphere in a 200 mL round-bottom flask. After 2.5 h reaction at 50 °C, the solution was precipitated into heptane with vigorous stirring. The solid product was collected by filtration and purified by recrystallization. Yield: 75%.

### Synthesis of poly(γ-benzyl-L-glutamate) (PBLG) terminated with vinyl group

BLG-NCA (3.0 g, 11.4 mmol) was dissolved in dried DMF (50 mL) at room temperature and then 3-buten-1-amine (42 μL, 0.46 mmol) was added to initiate the polymerization in nitrogen atmosphere. After 48 h reaction, acrylylchloride (187 μL, 2.3 mmol), DMAP (278 mg, 2.3 mmol) and HOBt (60 mg, 0.44 mmol) were added into the solution at 0 °C to couple the amino end with vinyl group. After 24 h reaction the solution was precipitated in diethyl ether. Finally, the solid product (2.5 g, 92% yield) was obtained after overnight drying under vacuum.

### Synthesis of peptide crosslinkers (PCs)

PBLG terminated with vinyl group (2.5 g) was dissolved in TFA (50 mL) and HBr (4 mL) at 0 °C to deprotect the glutamic acids. After 5 h reaction, the product was precipitated into diethyl ether and collected by centrifugation. The final product was purified by dialysis and lyophilized. Yield: 86%. The degree of polymerization (DP) of the PC was determined to be 22 and named as PC22. PC12 and PC32, with a DP of 12 and 32, respectively, were synthesized using the same procedure but different monomer/initiator ratio (Supplementary Table 1).

### Synthesis of the peptide-enhanced eutectogels

To prepare deep eutectic solvents (DES), ChCl (13.96 g, 0.1 mol) and EG (12.41 g, 0.1 mol) or urea (12.01 g, 0.1 mol) were mixed and stirred at 80 °C until clear liquids were obtained. To prepare eutectogels, PC22 (1.67 g, 0.563 mmol) and AAm (4.00 g, 56.3 mmol) were dissolved in DES (10.00 g), to which the photo-initiator DEAP (22.7 μL) was added at 0.2% of AAm molar content. The solution was poured into a PTFE mold and cured under UV light (320–420 nm, and 50 mW cm$^{-2}$) for 5 min. The resulting gel was named as $A_{0.4}PC22_{1\%}$ gel according to the composition of the pre-gel solution. Other gels were synthesized using the same procedure.

### 3D printing

3D-printed ink was identical to the pre-curing solution used as describe above. A self-built multi-axis micro-positioning system (Prusa i3) was used for 3D printing, and the nozzle was modified for piston extrusion. The inks were all placed in a 1 mL customized precision feeder with the nozzle was 410 μm. 3D printing speed measured in terms of printhead travel speed is 60 mm min$^{-1}$. During the printing process, a 60 W UV lamp was used to initiate polymerization and cross-linking. Cyclohexane was used as protective liquid and coagulation bath during printing processes for more regular structure.

### Mechanical experiments

All mechanical experiments were performed on a universal testing machine (CTM2100, Xie Qiang Instrument Manufacturing), except for the tensile tests at different temperatures which were measured on a force measurement sensor system (CY-GJD, Cheng Ying Instrument Manufacturing). If not otherwise specified, all tensile and compressive tests were performed at room temperature and a speed of 50 mm min$^{-1}$. The samples for tensile tests were made in a dumbbell shape of type III in accordance with GB/T528-2009 standard, which is equivalent to ISO 37:2005. The samples for compressive tests were made into cylinders with 8 mm height and 10 mm diameter. All tests were repeated at least four times.

### Adhesion strength

The adhesion strength of the gels was measured by lap shear tests (ASTM F2255) using the same machine for the mechanical tests at a speed of 10 mm min$^{-1}$. All gel samples had a size of 2.5 cm × 1.0 cm × 0.2 cm. Commercial glass slides were used as adherends. Each test was repeated 5 times. Final the adhesion strength was obtained by calculating the maximum force divided by the overlapping contact area.

### Fracture experiments

Fracture toughness of the eutectogels was gauged by single-edge cracking tests at a strain speed of 25 mm min$^{-1}$. The width of the samples was 5.0 mm with a 2.5 mm crack. A Canon 600D camera with a macro lens was used to record breakage situation. The fracture energy G was calculated using the following formula according to the compliance method of Greensmith et al.[40].

$$G = 6AE/\sqrt{\lambda} \tag{1}$$

where A is the strain energy density calculated from the area covered by the uncracked sample stress-strain curve from 0% to the critical stretch λ, and E is the initial crack length.

### Manufacturing and characterization of sensors

The strain/pressure sensors were assembled using $A_{0.4}PC22_{1\%}$ gel with [ChCl][EG] as solvent according to refs. 16,25. Resistance measurements were carried out with a LK2000 electrochemical workstation (Lanliko Chemical Electronic High Tech). The relative resistance change used for strain sensor was calculated using the following formula:

$$\Delta R/R_0 = (R - R_0)/R_0 \tag{2}$$

where $R_0$ and $R$ were the resistance of the sensor before and after deforming, respectively.

The relative current change used for pressure sensor was calculated using the following formula:

$$\Delta I/I_0 = (I - I_0)/I_0 \tag{3}$$

where $I_0$ and $I$ were the current of the sensor before and after deforming, respectively.

### Other characterizations

$^1$H NMR spectra were tested on a Bruker Avance 400 MHz NMR Spectrometer. GPC was measured on a Hitachi L-2490 using DMF as the eluent at 40 °C with flow rate is 1 mL min$^{-1}$ and polymethylmethacrylate (PMMA) was used as standard. Transmission visible spectroscopy was obtained by using a Shimadzu TCC-240A UV-Vis spectrophotometer. Circular dichroism (CD) was obtained using a circular dichroism spectrometer (BioLogic, MOS-450) with 150 W Xenon lamp. A home-made stretching device which can fit into the sample cell of the circular dichroism spectrometer was used to stretch the gel sample to a predetermined strain. The schematic diagram and the picture of the stretching device were shown in Supplementary Fig. 27. DSC curve was measured on a Netzsch DSC204 thermal analysis system with the heating rate was 1 °C min$^{-1}$. The human tests were approved by the Medical Ethics Committee of Nankai University (No. 2020059). Informed consent was obtained from all participants.

## Data availability

The data that supports the findings of this study are available within the article and the Supporting Information of the article.

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

## Acknowledgements

The authors thank financial support for this work from the National Natural Science Foundation of China (Grants No: 52073146 (Y.G.), 52033004 (Y.Z.), 51873091 (Y.G.), and 52273109 (Y.Z.)). Yan Zhang thanks Dr. Rui Liu for his kind help in peptide synthesis.

## Author contributions

Yongjun Zhang and Ying Guan conceived and supervised the research. Yan Zhang and Yafei Wang performed the sample fabrication. Yan Zhang carried out the mechanical experiment, characterization and sensor test. Yongjun Zhang and Yan Zhang analyzed data and wrote the manuscript. All authors participated in discussions of the research.

## Competing interests

The authors declare no competing interests.
