## [Peer Review File · Nature Communications]

Peptide-enhanced tough, resilient and adhesive eutectogels for highly reliable strain/pressure sensing under extreme conditionsReviewers' Comments:

Reviewer #1:

Remarks to the Author:

This work describes the fabrication of a polyacrylamide gel with a peptide crosslinker and swollen in a deep eutectic solvent. The mechanical properties of this gel are characterized, along with the adhesive properties, and its application as a strain sensor. Ultimately I do not feel this paper is suitable for publication in Nature Communications.

1) Hydrogels with the same crosslinker have been previously published, and I don't believe switching to a eutectic solvent represents sufficient novelty, especially when similar good properties have been attained in hydrogels by using simpler techniques.

2) The mechanical properties of the gel are good but not great. Other gels exist that have similar or better properties. The hydrogels produced by He and coworkers (published in Nature last year) have much better mechanical properties from a "single network" gel. In general, I don't think it makes sense to separate the comparison of single and double network gels; the number of networks is relatively unimportant, except that it causes more complicated fabrication for composite applications.

3) The reason why these gels are tough is not rigorously explained, and the current discussion about peptide chains extending and "folding back" feels extremely qualitative. I think for this work to be considered, a deeper explanation of the process by which the gels achieve improved properties is required. As currently written, the paper mainly explains the results of the measurements.

4) The adhesion tests are not done well. Reporting the stress of a lap-shear test has no meaning, because lap shear joints are highly dependent on the sample thickness. A poor adhesive that is thin will achieve higher strength than a thickly applied "strong" adhesive. If you want to use lap shear tests to report adhesion data, interfacial G_c values at rates approaching zero ($G_c, 0$) should be reported.

5) the authors highlight the anti-freezing and anti-drying properties of these materials. I think this is a benefit of using eutectic solvents, but I don't think this is a ground-breaking finding. Suo and coworkers swelled hydrogels in varying types of salts in order to prevent them from drying, and would also lower the freezing point. The incorporation of salts would also make the hydrogels conductive, another point that is stressed in this paper. How are DES gels better than saline-swollen hydrogels?

6) The "preliminary biocompatibility" findings (that the gel didn't cause itching or other allergic reactions) seem anecdotal and somewhat problematic. I don't think it is good to test on "volunteers" especially if there aren't human testing protocols listed. A robotic finger would work just as well.

7) there are numerous spelling mistakes throughout the paper.

Reviewer #2:

Remarks to the Author:

In this contribution, the authors developed a series of eutectogels composed of crosslinked polyacrylamide and choline chloride-based deep eutectic solvents. The free-radical polymerization of acrylic monomers in DESs is a strategy often used to create polymer networks in situ, called eutectogels. The approach presented here is relevant for the development of materials for soft robotics and flexible electronics. The main novelty relies on the introduction of acrylic-modified peptides as the crosslinker of the polyacrylamide network, which is an interesting mechanism of energy dissipation during gel deformation in DES, as opposed to a recently published work (Chem. Eng. J. 2021, 415, 128839) where the same authors reported the same system of polyacrylamide

crosslinked by the same peptide but in water (hydrogel). Two salient results make this work relevant in the area:

- 1) the introduction of peptides, whose intramolecular hydrogen bonds due to a possible α -helix conformation provide a new energy dissipation strategy for eutectogels, and
- 2) the long-term signal stability collected upon the eutectogel deformation at extreme temperature conditions and for prolonged times.

The structure-property relationship of the eutectogels was investigated using standard techniques aimed to demonstrate the chemical structure of some of the eutectogels ingredients, e.g., modified peptide by NMR, GPC, and the whole eutectogel by DSC, and how these components impact the mechanical properties which were thoroughly investigated from -20 to 80 Celcius.

Overall, the eutectogel performance and salient properties place this work above many similar published materials. However, this reviewer suggests that aspects of the investigation, mainly concerning additional chemical characterizations, need to be improved before the manuscript becomes accepted for Nature Communications. Specific comments and suggestions are as follow:

1. In the abstract, line 28, please specify what "extreme conditions" refer to (temperature, pressure, saline concentration, etc.).
2. In the introduction, lines 84-85 "Thanks to the good conductivity of DESs, the resulting eutectogels themselves are conductive without the need to add any other substances." add appropriate references such as those reviewed in J. Phys. Chem. B 2020, 124, 39, 8465–8478 & Matter 2021, 4, 7, 2141-2162
3. Supplementary Figures 2-5 show H NMR spectra of only one modified peptide (PC12, PC22, or PC32). The authors should add the spectra of the two other modified peptides. Is it possible to note the differences in the integration of the acrylic moieties relative to the peptide repeating unit by H NMR?
4. For the data in Supplementary Table 1, what PC was used to assess the dissolution of the PC in DESs? This needs to be indicated in the caption. Is there any difference in solubility between the peptides with different MW?
5. Add specifications about the photoinitiation of the free radical polymerization. For instance, the wavelength used in the UV lamp?
6. MAJOR. What is the conversion of monomer to polymer? Are there any monomers traces remaining in the eutectogel? Consider that acrylamide monomer is regarded as a toxic substance, and exposure to the monomer increases the risk for several types of cancer in humans. In the same line, was the work revised by an appropriate instance for using the materials in humans?
7. MAJOR. Chemical characterization of the polymer network formed in situ in the DEs is absent. The authors are encouraged to wash out the DES and characterize the polymeric network by suitable spectroscopies (e.g., FTIR, solid-state NMR) and thermal methods (e.g., DSC, TGA) and compare it with polyacrylamide (without peptide-crosslinking) to get insights into their chemical structure and disregard the presence of unreacted double bonds, for instance.
8. About the ink for 3D printing, how stable is the reactive ink? Free-radical polymerization is known to occur, triggered by ambient light and by self free-radical polymerization of monomers possibly catalyzed by the DES components (Prog. Polym. Sci. 2018, 78, 139-153). The viscosity of the whole reaction mixture needs to be evaluated. This will help the authors reach conclusions about the inks' stability and processability during 3D printing. What is the spacial resolution of the 3D printed structures?
9. What is the moisture content in the eutectogel, considering that many choline chloride-based DES are highly hygroscopic.
10. MAJOR. It is claimed that the peptide's helical structure is destructed during the mechanical deformation and that such an event is confirmed by Circular Dichroism. This needs to be discussed in detail as it is the foundation of the excellent performance of the eutectogel. The exact procedure to acquire the spectra of samples before stretching, being stretched at 1500% strain and unloaded, needs to be added and carefully discussed. What is the behavior of the peptide alone in DES by CD? Does it correlate with that seen when taking part in the polymer network in the eutectogel?

Sincerely,

Josué D. Mota-Morales

Reviewer #3:

Remarks to the Author:

The paper deals with the preparation of peptide-based polymeric eutectogels and their use as strain/pressure sensors.

The manuscript shows a high level of novelty for such kind of materials. Indeed, as stated by the AA, such kind of systems are frequently prepared in water. The manuscript is well written and properties as well as applications of the materials obtained are deeply analysed.

As a consequence, conclusions are well supported by experimental data.

For all above reasons, this referee thinks that the manuscript is suitable of publication in the present form.

Reviewer #4:

Remarks to the Author:

The manuscript by Zhang and co-workers details the preparation of polymer gels prepared from a deep eutectic solvent (DES) where an alpha-helical peptide is embedded into the structure. The resulting gel has impressive tensile and compressive strength, fracture toughness, fatigue resistance and adhesion. The authors show excellent repeatability with respect to sample deformation over 10000 cycles. The use of a deep eutectic solvent confers properties on the gel that are not normally accessible compared to those prepared in water, including use in sub-zero temperature conditions.. These materials have potential for use as conductive, flexible strain sensors.

The process is driven by a cross-linker that consists of a di-functional polypeptide consisting of polymerizable vinyl groups at each terminus, in the UV-mediated polymerization of an acrylamide matrix from two different Type 3 DES systems (choline chloride/urea and choline chloride/ethylene glycol). The resulting gels are highly transparent and stretchy, and can be used for 3D printing. The mechanical performance of these gels is significantly enhanced compared to a traditional cross linker (bis-acrylamide). The gels are tested in the context of strain sensors and pressure sensors with good performance. The gels are remarkably good at energy dissipation, which is attributed to the structure of the peptide cross-linker.

This is an impressive piece of work and the benchmarking of performance relevant to other reported eutectic-derived gels. Based on the Supporting Information this gel has the 2nd highest resilience, 4th highest toughness and 2nd highest fracture energy of the 9 reported. The elongation and strength at break are also very good but probably mid-range with respect to other DES-based systems; however importantly one does not trade off one category for another. The simplicity of synthesis, and convenience of using deep eutectic solvents makes this a notable contribution.

I have some specific points to be raised to the authors regarding questions that I feel are unresolved in the text and should be considered.

- The peptide cross-linker (PC) has two vinyl groups of differing reactivity (an allyl terminus and an acrylate terminus). These are likely to have different reactivity ratios in the copolymerization of acrylamide which possesses an electron withdrawing group. Has this been confirmed - for example are there unreacted allyl pendants (for example) present within the final structure after photopolymerization?

- Acrylamide itself has the potential to participate into the eutectic structure given its capacity to H-bond. Has this been examined, for example by 1D/2D NMR of the eutectic with acrylamide included? I anticipate that the authors have created a ternary eutectic (quaternary if you consider the PC).
- In the text (lines 159-168), the authors refer to changing the PC loading, but are referencing the wrong figures (this should be Figs 3e /3f).
- Figure 5 - what is the mass of the eutectogel in panel a? It would be interesting to know the mass/mass ratio that these gels can support via adhesion
- In all photos of gels (e.g. Fig 6f and similar), please provide a scale bar for reference.
- I am confused by the DSC trace in Figure 6c. Firstly, is this a heating or cooling trace? Secondly, provide the trace in the opposite direction. Were multiple heating/cooling loops performed? An endothermic transition is visible, can the authors describe this (is it a melt transition?) DSC hysteresis is often seen in deep eutectic solvents and is worth discussing. The authors discuss that the gel freezing point is -54.4 deg C (line 284), however this temperature is not even in the scale of the figure.
- One thing standing out for me is that these eutectogels can potentially be purified (the EG/urea and choline chloride can be removed via washing) and the resulting cross-linked network can be re-dispersed/hydrated with water. Based on the results presented, this gel should show inferior performance. It would however be an interesting reference point in comparison to the gels prepared directly in water (e.g. the data in Figure 5b). I raise this, as the high viscosity of the eutectic changes the nature of the photopolymerization process and thus cross-linking density and ultimately mechanical response.

Point-to-point response to reviewers' comments

We are very grateful for the reviewers' constructive comments and suggestions. The following are our point-to-point responses to their comments:

Response to Reviewer #1's comments:

Reviewer #1 (Remarks to the Author):

This work describes the fabrication of a polyacrylamide gel with a peptide crosslinker and swollen in a deep eutectic solvent. The mechanical properties of this gel are characterized, along with the adhesive properties, and its application as a strain sensor. Ultimately I do not feel this paper is suitable for publication in Nature Communications.

1) Hydrogels with the same crosslinker have been previously published, and I don't believe switching to a eutectic solvent represents sufficient novelty, especially when similar good properties have been attained in hydrogels by using simpler techniques.

Answer: As we pointed out in the Introduction, although hydrogels with significantly improved toughness, resilience, adhesion, anti-freezing and anti-drying properties have been reported, the synthesis of a gel with all these properties remains a huge challenge. Here by combining peptide crosslinker strategy and the deep eutectic solvent strategy, we successfully overcome this problem. The resulting gels exhibit excellent toughness, resilience, adhesion, anti-freezing and anti-drying properties simultaneously. As commented by Reviewer 4, "This is an impressive piece of work and the benchmarking of performance relevant to other reported eutectic-derived gels."

It is true we previously used the same peptide crosslinker to synthesize hydrogels. Here the peptide crosslinker strategy was combined with the deep eutectic solvent strategy to synthesize gels with excellent overall performance. In fact, the combination of two or more strategies is a commonly used and efficient way to get a solution for hard problems. A good example is the Nature paper by He and coworkers the reviewer mentioned later. (Nature, 2021, 590, 594) In Nature paper the authors synthesized strong tough hydrogels by combining directional freeze-casting and a subsequent salting-out treatment. Both the directional freeze-casting and the salting-out strategies have been extensively used previously. However, we still believe this work is a great piece of work.

The reviewer claimed that "similar good properties have been attained in hydrogels by using simpler techniques" but failed to give an example to support this proposition. To the best of our knowledge, this is not the fact. Gels with overall performance, i.e., with excellent toughness, resilience, adhesion, anti-freezing and anti-drying properties simultaneously, as good as our gels, have not been reported yet. In fact our method is already a very simple one.

2) The mechanical properties of the gel are good but not great. Other gels exist that have similar or better properties. The hydrogels produced by He and coworkers (published in Nature last year) have much better mechanical properties from a "single network" gel. In general, I don't think it makes

sense to separate the comparison of single and double network gels; the number of networks is relatively unimportant, except that it causes more complicated fabrication for composite applications.

Answer: As we pointed out in the Introduction, our purpose is to synthesize gels with excellent overall performance, i.e., gels with excellent toughness, resilience, adhesion, anti-freezing and anti-drying properties simultaneously. It is true that many gels with excellent toughness, or excellent resilience, or excellent adhesion, or excellent anti-freezing, or excellent anti-drying properties have been reported. Particularly He and coworkers synthesized hydrogels with good mechanical properties. The tensile strength, elongation at break and toughness of the hydrogels were very high. However, their gels failed to exhibit good resilience, adhesion, anti-freezing and anti-drying properties. For example, as Fig 4b in their paper shows (see below), very large hysteresis loops were observed when cyclic tensile tests were performed, indicating low resilience of the gels.

Fig 4b. Ximin He et al., Nature, 2021, 590, 594

Very differently, very small hysteresis loops were observed when our gels were subjected to consecutive loading-unloading cycles, indicating high resilience of the gels:

Fig. 3g in this manuscript

Besides excellent toughness and resilience, our gels also exhibit excellent adhesion, anti-freezing and anti-drying properties. We would like to note that, for the real applications of gels, it is more important to improve the overall performance of the gel, rather than just one single property.

Thanks to the excellent overall performance of the gel, unprecedented long-term signal stability was achieved when it was used as strain sensor. In contrast, for gels with excellent mechanical properties but without properties such as high resilience and anti-drying, it is impossible to achieve this goal.

3) The reason why these gels are tough is not rigorously explained, and the current discussion about peptide chains extending and “folding back” feels extremely qualitative. I think for this work to be considered, a deeper explanation of the process by which the gels achieve improved properties is required. As currently written, the paper mainly explains the results of the measurements.

Answer: In the 3rd paragraph on Page 7 we explained why the mechanical properties of the PC-crosslinked eutectogels are significantly improved. Two reasons were given. One is that the incorporation of long peptide chains into the network reduces the inhomogeneity of the network. Another reason is that the breakage of the intramolecular hydrogen bonds stabilizing the α -helical structure of the peptide chains provides a novel mechanism for energy dissipation. In our previous paper (Ref 25) these reasons have been explained in details. Therefore in this manuscript, these reasons were explained in a relatively brief manner.

According to the reviewer’s suggestion, the CD spectra of the peptide crosslinker dissolved in DESs were also measured. The results indicate the peptide crosslinker can maintain its α -helical structure in DESs.(Supplementary Figure 14) The stretching device for the measurement of CD spectra of the stretched samples was added. (Supplementary Figure 27) In addition, the second reason, i.e., the breakage of intramolecular hydrogen bonds for energy dissipation, was revised as follows: “As revealed by CD spectra (Supplementary Figure 14) the peptide crosslinker dissolved in the DESs adopts an α -helical structure, which is evidenced by the two minima at 213 and 225 nm. More importantly the peptide chains incorporated in the eutectogels also adopt an α -helical structure. (Supplementary Figure 14 and 15) It is well-known that the α -helical structure is stabilized by hydrogen bonds between the C = O group of each amino acid and the NH group of amino acid four residues earlier in the sequence³³. These intramolecular hydrogen bonds can be fractured by external force, and thus dissipating energy applied on the gel³⁴. According to previous AFM study, the energy required to break the intramolecular hydrogen bonds is 20.2 kJ/mol³⁴. The breakage of the intramolecular hydrogen bonds leads to the destruction of the α -helical structure, which was evidenced by significant change in the CD spectra of the stretched gel (Supplementary Figure 15).”

4) The adhesion tests are not done well. Reporting the stress of a lap-shear test has no meaning, because lap shear joints are highly dependent on the sample thickness. A poor adhesive that is thin will achieve higher strength than a thickly applied “strong” adhesive. If you want to use lap shear tests to report adhesion data, interfacial Gc values at rates approaching zero (Gc,0) should be reported.

Answer: The adhesion properties of the gels were measured using the standard lap-shear test (ASTM F2255), in which all the samples had the same size (2.5 cm × 1.0 cm × 0.2 cm). We understand the sample thickness plays a key role in the measurement of adhesion. All the samples had the same thickness.(0.2 cm) In fact, the same method has been widely used to measure adhesion properties of the gels in the literature, e.g. Nature Communications, 2021, 12, 1670; Nature Biomedical

Engineering, 2021, 5(10): 1131-1142; Nature Materials, 2021, 20(2): 229-236); ACS Nano, 2020, 14(12): 17004-17017; Bioactive Materials 13 (2022) 260–268; Nat. Biomed. Eng (2022). <https://doi.org/10.1038/s41551-022-00905-2>

In the revised manuscript, to add more details of the lap-shear test particularly the size of the samples, the sentence “The adhesion strength of the gels was measured by lap shear tests using the same machine for the mechanical tests at a speed of 10 mm min⁻¹.” was revised to be “The adhesion strength of the gels was measured by lap shear tests (ASTM F2255) using the same machine for the mechanical tests at a speed of 10 mm min⁻¹. All gel samples had a size of 2.5 cm×1.0 cm× 0.2 cm.” (Page 4 Line 4 from bottom, Supplementary information)

5) the authors highlight the anti-freezing and anti-drying properties of these materials. I think this is a benefit of using eutectic solvents, but I don't think this is a ground-breaking finding. Suo and coworkers swelled hydrogels in varying types of salts in order to prevent them from drying, and would also lower the freezing point. The incorporation of salts would also make the hydrogels conductive, another point that is stressed in this paper. How are DES gels better than saline-swollen hydrogels?

Answer: Again we would like to emphasize that our purpose is to synthesize gels with excellent overall performance, i.e., gels with excellent toughness, resilience, adhesion, anti-freezing and anti-drying properties simultaneously. As we mentioned before, hydrogels with significantly improved toughness, resilience, adhesion, anti-freezing and anti-drying properties have been reported, however, the synthesis of a gel with all these properties remains a huge challenge.

As the reviewer mentioned, Suo and coworkers swelled hydrogels in varying types of salts in order to prevent them from drying, and would also lower the freezing point. The incorporation of salts would also make the hydrogels conductive. (Applied Physics Letters, 2014, 105(15), 151903) However, this work only improved the water retention capacity of the hydrogel. Other properties, such as mechanical properties and adhesion properties, were not improved.

It is a good idea to enhance water retention capacity of a hydrogel by introducing highly hydratable salts, as Suo et al demonstrated. However, by comparing the results of Suo et al with ours, we can conclude that the deep eutectic solvent strategy can get much better anti-drying properties. As the Fig. 2 of the Suo article (Applied Physics Letters, 2014, 105(15): 151903) (see below) shows, although the introduction of highly hydratable salts slows down water loss, a significant water loss still occurs within 48 h at temperature of 25 °C and relative humidity of 30% RH. Specially, polyacrylamide hydrogel containing high content of lithium chloride can only retain over 70% of its initial water at a relative humidity of 10% RH.

Fig.2. Zhigang Suo et al, Applied Physics Letters, 2014, 105(15): 151903. Evolutions of water loss with time for different hydrogels kept in chamber with temperature of 25 °C and relative humidity of 30% RH.

Very differently, because of the extremely low vapor pressure of deep eutectic solvents, our eutectogels exhibit much better anti-drying properties. After 30 days of storage under ambient environmental conditions, the mass loss is as small as only 1.4% (Fig. 6d, see below):

Fig. 6d in this manuscript. Weighs of a peptide-enhanced eutectogel and a hydrogel stored under ambient conditions.

In the revised manuscript, the Suo article (Applied Physics Letters, 2014, 105(15): 151903) was added as a new reference (Ref 24), to honor their achievement in development hydrogels with anti-drying properties.

6) The “preliminary biocompatibility” findings (that the gel didn’t cause itching or other allergic reactions) seem anecdotal and somewhat problematic. I don’t think it is good to test on “volunteers” especially if there aren’t human testing protocols listed. A robotic finger would work just as well.

Answer: As a preliminary test, we attached the gels on the skin of a volunteer. After 16 h wearing, no itching or other allergic reactions were observed, suggesting good biocompatibility of the gel. The volunteer was the first author of the manuscript. Before human tests, the testing protocol was checked and approved by the Medical Ethics Committee of Nankai University (No. 2020059).

In fact, the same method was widely used in the literature. The followings are some examples: Nature communications, 2020, 11(1): 1-13, Chemical Engineering Journal, 2022: 137878, and npj Flexible Electronics (2021) 5:23.

In the revised manuscript, the sentence “The human tests were approved by the Medical Ethics Committee of Nankai University (No. 2020059).” was added at the end of the “Other characterizations” section in Supplementary information.

The following references were added as Ref 46, 52, and 53 on Page 16 Line 9 from bottom: Nature communications, 2020, 11(1): 1-13, Chemical Engineering Journal, 2022: 137878, and npj Flexible Electronics (2021) 5:23.

7) there are numerous spelling mistakes throughout the paper.

Answer: Thanks for pointing out this problem. We carefully checked the manuscript again. Spelling mistakes and other language problems were corrected.

Response to Reviewer #2's comments:

Reviewer #2 (Remarks to the Author):

In this contribution, the authors developed a series of eutectogels composed of crosslinked polyacrylamide and choline chloride-based deep eutectic solvents. The free-radical polymerization of acrylic monomers in DESs is a strategy often used to create polymer networks in situ, called eutectogels. The approach presented here is relevant for the development of materials for soft robotics and flexible electronics. The main novelty relies on the introduction of acrylic-modified peptides as the crosslinker of the polyacrylamide network, which is an interesting mechanism of energy dissipation during gel deformation in DES, as opposed to a recently published work (Chem. Eng. J. 2021, 415, 128839) where the same authors reported the same system of polyacrylamide crosslinked by the same peptide but in water (hydrogel). Two salient results make this work relevant in the area:

- 1) the introduction of peptides, whose intramolecular hydrogen bonds due to a possible α -helix conformation provide a new energy dissipation strategy for eutectogels, and
- 2) the long-term signal stability collected upon the eutectogel deformation at extreme temperature conditions and for prolonged times.

The structure-property relationship of the eutectogels was investigated using standard techniques aimed to demonstrate the chemical structure of some of the eutectogels ingredients, e.g., modified peptide by NMR, GPC, and the whole eutectogel by DSC, and how these components impact the mechanical properties which were thoroughly investigated from -20 to 80 Celcius.

Overall, the eutectogel performance and salient properties place this work above many similar published materials. However, this reviewer suggests that aspects of the investigation, mainly concerning additional chemical characterizations, need to be improved before the manuscript becomes accepted for Nature Communications. Specific comments and suggestions are as follow:

1. In the abstract, line 28, please specify what "extreme conditions" refer to (temperature, pressure, saline concentration, etc.).

Answer: Many thanks for the suggestion. Here "extreme conditions" mainly refer to sub-zero temperatures and long-term application. In the revised manuscript, the sentence "However, nearly every natural biological structure relies on water as solvents or carriers, which limits the possibility in extreme conditions." was revised to be "However, nearly every natural biological structure relies on water as solvents or carriers, which limits the possibility in extreme conditions, such as sub-zero

temperatures and long-term application.”. (Page 2 Line 3)

2. In the introduction, lines 84-85 "Thanks to the good conductivity of DESs, the resulting eutectogels themselves are conductive without the need to add any other substances." add appropriate references such as those reviewed in J. Phys. Chem. B 2020, 124, 39, 8465–8478 & Matter 2021, 4, 7, 2141-2162

Answer: Many thanks for the informative advice. It is necessary to include relevant literatures to support the high conductivity of deep eutectic solvents. Both references the reviewer mentioned are very appropriate for this purpose. In the revised manuscript the two references were added as Ref 27 and 28. (Page 4 Line 6 from bottom)

3. Supplementary Figures 2-5 show ¹H NMR spectra of only one modified peptide (PC12, PC22, or PC32). The authors should add the spectra of the two other modified peptides. Is it possible to note the differences in the integration of the acrylic moieties relative to the peptide repeating unit by ¹H NMR?

Answer: Many thanks for these suggestions. According to the suggestion, the ¹H NMR spectra of the PBLG precursors of PC22 and PC 32 were added in Supplementary Figure 3. The ¹H NMR spectra of the acryl-capped PBLG precursors of PC22 and PC 32 were added in Supplementary Figure 4. The ¹H NMR spectra of peptide crosslinkers PC22 and PC 32 were added in Supplementary Figure 5.

According to the reviewer’s suggestion, the degree of polymerizations (DPs) of the peptide crosslinkers were also calculated from the ratio of the integration of the acrylic moieties and the peptide repeating unit. The results were added in Supplementary Table 1. The DPs calculated from ¹H NMR spectra are close to the ones calculated from GPC. From GPC, the DPs were determined to be 12.4, 22.2 and 32.1, while they were determined to be 11.6, 21.2 and 31.0 from ¹H NMR spectra.

4. For the data in Supplementary Table 1, what PC was used to assess the dissolution of the PC in DESs? This needs to be indicated in the caption. Is there any difference in solubility between the peptides with different MW?

Answer: Many thanks for pointing this problem. The PC used to evaluate the dissolution in DESs was PC22, i.e., the PC with a degree of polymerization of 22. To specify the PC used for the evaluation, in the revised manuscript, the caption of the table was revised to be “Dissolution of PC22 in DESs.”.

To study the effect of MW on the solubility of the peptide crosslinkers, the solubilities of the three PCs in [ChCl][EG] were measured. The results were added as a new figure in the revised supporting information (Supplementary Figure 6) (also shown below). With increasing MW of the peptide crosslinker, the solubility decreases slightly. In the revised manuscript, a brief discussion was added as “All 3 PCs dissolve well in the solvents, but the solubility decreases slightly with increasing molecular weight.(Supplementary Figure 6)” (Page 5 Line 2 from bottom)

Supplementary Figure 6 | Solubility of the PCs with different DPs in [ChCl][EG].

5. Add specifications about the photoinitiation of the free radical polymerization. For instance, the wavelength used in the UV lamp?

Answer: Many thanks for the suggestion. According to the reviewer's suggestion, in the revised manuscript, specifications about the photoinitiated polymerization were added, including wavelength and intensity of the UV lamp.(320-420 nm, and 50 mW cm⁻²) (Page 3 Line 5 from bottom in the Supplementary information)

6. MAJOR. What is the conversion of monomer to polymer? Are there any monomers traces remaining in the eutectogel? Consider that acrylamide monomer is regarded as a toxic substance, and exposure to the monomer increases the risk for several types of cancer in humans. In the same line, was the work revised by an appropriate instance for using the materials in humans?

Answer: Many thanks for raising this important question. It is noteworthy that many polyacrylamide gels were synthesized in the literature by free-radical polymerization of AAm and then used for wearable sensors, however, no one measured the AAm content remaining in the gels and considered its potential hazard.

To answer the question, we measured the AAm contents remaining in the gels by HPLC, and found it was 0.0359 wt%. The conversion rate of the AAm monomer was thus determined to be 99.86%. The results indicate a high conversion rate of the AAm monomer.

Although the conversion rate of the AAm monomer is high, there is still a small amount of AAm left in the gels. As the reviewer pointed out, AAm is toxic. In fact, the toxicity of AAm has been extensively investigated. A. Shipp et al reviewed the toxicity data and gave a reference dose (RfD) of 0.83 µg/kg/day based on reproductive effects, and 1.2 µg/kg/day based on neurotoxicity. (Critical Reviews in Toxicology, 2006, 36(6-7), 481-608) Roughly, if one exposes to AAm chronically, a daily dose below 1 µg/kg can be regarded safe.

The results of A. Shipp et al were used to judge if it is safe for a human being with a body

weight of 50 kg to wear 1g of gel for 1 day. The total amount of AAm in the gel is 359 μg . Also from the Review by A. Shipp et al we know that the dermal absorption of AAm ranges from 25% to 29% of the amount applied daily. Therefore the amount of AAm absorbed by the people will be $\sim 90 \mu\text{g}$, corresponding to a daily dose of $1.8 \mu\text{g}/\text{kg}$. This value is larger than the reference dose given by A. Shipp et al ($1 \mu\text{g}/\text{kg}$), indicating chronic wearing of the gel may cause health problems. However, since the reference dose is intended for chronic exposure, short-term wearing of the gels, for example 24 h, can be regarded safe. Even so we think it is better to further reduce the residual AAm content in the gel, or replace it with a less toxic monomer.

In the revised manuscript, the following sentences were added on Page 6 Line 7: “The AAm contents remaining in the gels was measured by HPLC to be 0.0359 wt%. The conversion rate of the AAm monomer was thus determined to be 99.86%.”.

7. MAJOR. Chemical characterization of the polymer network formed in situ in the DEs is absent. The authors are encouraged to wash out the DES and characterize the polymeric network by suitable spectroscopies (e.g., FTIR, solid-state NMR) and thermal methods (e.g., DSC, TGA) and compare it with polyacrylamide (without peptide-crosslinking) to get insights into their chemical structure and disregard the presence of unreacted double bonds, for instance.

Answer: Many thanks for pointing out the problem. According to the reviewer’s suggestion, PC22-crosslinker eutectogel samples and BIS-crosslinked eutectogel samples were synthesized. The samples were repeatedly soaked in deionized water to wash out the DES. They were then dried and their FTIR and solid-state NMR spectra and DSC and TGA thermograms were measured. These results were added as Supplementary Figure 8-11 in the revised manuscript and also shown below.

Supplementary Figure 8| FTIR spectra of A_{0.4}PC22_{1%} and A_{0.4}BIS_{1%} gels. The gels were synthesized in [ChCl][EG]. Before measurement, the gels were washed with DI water to remove the DES and dried.

Supplementary Figure 9| ^{13}C solid-state NMR spectra of $\text{A}_{0.4}\text{PC}_{22}1\%$ (a) and $\text{A}_{0.4}\text{BIS}_{1\%}$ gels (b). The gels were synthesized in $[\text{ChCl}][\text{EG}]$. Before measurement, the gels were washed with DI water to remove the DES and dried.

Supplementary Figure 10| DSC thermograms of $\text{A}_{0.4}\text{PC}_{22}1\%$ (a) and $\text{A}_{0.4}\text{BIS}_{1\%}$ gels (b). The gels were synthesized in $[\text{ChCl}][\text{EG}]$. Before measurement, the gels were washed with DI water to remove the DES and dried.

Supplementary Figure 11| TGA of A_{0.4}PC22_{1%} and A_{0.4}BIS_{1%} gels. The gels were synthesized in [ChCl][EG]. Before measurement, the gels were washed with DI water to remove the DES and dried.

Both the FTIR spectra and ¹³C solid-state NMR spectra of A_{0.4}PC22_{1%} are similar to that of A_{0.4}BIS_{1%}, suggesting similar chemical structure of the two gels. Because of the introduction of poly(glutamic acid) peptide chains, a shoulder was found at 1738 cm⁻¹ in the FTIR spectra of A_{0.4}PC22_{1%}, which was assigned to the carboxylic acid groups in the peptide chains. In addition, a peak appeared at 53 ppm on the ¹³C solid-state NMR spectra of A_{0.4}PC22_{1%}, which was assigned to the α-C on peptide chains.

From the FTIR spectra of the two gels, no signals of vinyl groups were found. Also no signals were found for vinyl groups from 100 to 150 ppm on their ¹³C solid-state NMR spectra. These results indicate that the amounts of unreacted double bonds left in the gels were negligible.

The thermal behaviors of the two gels are generally similar, but some differences were also observed. The DSC study reveals that the introduction of peptide chains into the gels lowers the glass transition temperature from 92°C to 57°C. The result may suggest that the peptide chains acted as plasticizer for the polyacrylamide chains. TGA study reveals that the introduction of peptide chains lowered the thermal stability of the gel.

Besides adding the new figures in Supporting Information, a brief discussion about the results was added as follows in the revised manuscript: “FTIR and ¹³C solid-state NMR studies revealed a similar chemical structure of peptide-crosslinked gels with the BIS-crosslinked ones. (Supplementary Figure 8 and 9) The amount of unreacted double bonds left in the gels was negligible. The peptide-crosslinked gels also present thermal behaviors similar to that of BIS-crosslinked gels, as demonstrated by DSC and TGA studies. (Supplementary Figure 10 and 11)” (Page 6 Line 9)

8. About the ink for 3D printing, how stable is the reactive ink? Free-radical polymerization is known to occur, triggered by ambient light and by self free-radical polymerization of monomers

possibly catalyzed by the DES components (Prog. Polym. Sci. 2018, 78, 139-153). The viscosity of the whole reaction mixture needs to be evaluated. This will help the authors reach conclusions about the inks' stability and processability during 3D printing. What is the spacial resolution of the 3D printed structures?

Answer: Many thanks for the good question. To study the stability of the pre-gel solutions, they were stored for 7 days at 4 °C in a refrigerator. No obvious change was observed. The result indicates that the pre-gel solutions are quite stable. The images of freshly prepared pre-gel solutions and the solution after 7 days storage at 4 °C were added as Supplementary Figure 12 and also shown below:

Supplementary Figure 12| Images of the pre-gel solution of $A_{0.4}PC_{22}1\%$ [ChCl][EG]. (a) Just prepared. (b) After 7 days storage in a refrigerator at 4 °C.

The viscosity of the pre-gel solution was determined to be ~ 175 mPa s. According to a new review by Puza and Lienkamp, for successful extrusion 3D printing, the hydrogel precursor inks or melts need a viscosity in the range of 6 to 30×10^7 mPa s. (Advanced Functional Materials, 2022. n/a(n/a): p. 2205345.) Therefore the viscosity of the pre-gel solution is suitable for 3D imprinting. We also monitored the viscosity of the pre-gel solution during 7 days storage. No significant change was observed, indicating high stability of the solution under the storage conditions. The viscosity of the pre-gel solution of $A_{0.4}PC_{22}1\%$ [ChCl][EG] during 7 days storage was added as Supplementary Figure 13| and also shown below:

Supplementary Figure 13| Viscosity of the pre-gel solution of $A_{0.4}PC_{22}1\%$ [ChCl][EG] during 7

days storage in a refrigerator at 4 °C.

As for the spacial resolution of the 3D printed structures, the nozzle we used is 410 µm. The layer thickness is 100 µm. The positioning accuracy of the XY plane is 10 µm, and the positioning accuracy of the Z axis is 10 µm.

In the revised manuscript, the reference (Prog. Polym. Sci. 2018, 78, 139-153), a good review about free-radical polymerizations of and in deep eutectic solvents, was added as Ref 29 when discussing the gel synthesis by polymerization. (Page 6 Line 3)

The above figures were added as Supplementary Figure 12 and 13.

The part discussion about 3D printing was revised to be: “As 3D printing ink, the viscosity of the pre-gel solutions was determined to be ~175 mPa s, making them suitable ink for 3D printing.³¹ They are also stable when stored at 4°C in a refrigerator.(Supplementary Figure 12 and 13) The resolution of the resulting 3D structure is 410 µm in x direction and 100 µm in y direction.” (Page 6 Line 4 from bottom)

9. What is the moisture content in the eutectogel, considering that many choline chloride-based DES are highly hygroscopic.

Answer: Thanks for the question. The water content in the was measured by Karl-Fischer coulometric titrations. As the reviewer anticipated, the eutectogel contains a small amount of water, because the DESs are hygroscopic. The water content of the A_{0.4}PC22_{1%} [ChCl][EG] gel was determined to be 2.77%.

In the revised manuscript, the following sentence was added on Page 6 Line 10 from bottom: “The moisture content in the eutectogel was determined to be 2.77% by Karl-Fischer coulometric titrations.³⁰”

10. MAJOR. It is claimed that the peptide's helical structure is destructed during the mechanical deformation and that such an event is confirmed by Circular Dichroism. This needs to be discussed in detail as it is the foundation of the excellent performance of the eutectogel. The exact procedure to acquire the spectra of samples before stretching, being stretched at 1500% strain and unloaded, needs to be added and carefully discussed. What is the behavior of the peptide alone in DES by CD? Does it correlate with that seen when taking part in the polymer network in the eutectogel?

Answer: Many thanks for the suggestions. According to the reviewer's suggestions, we first measured the CD spectra of peptide crosslinker PC22 in [ChCl][EG] and [ChCl][Urea]. The results were added as Supplementary Figure 14 in the revised manuscript and also shown below:

Supplementary Figure 14 | CD spectra of PC22 solution in deep eutectic solvent and $A_{0.4}PC22_{1\%}$ gel in the same deep eutectic solvent. The deep eutectic solvent is [ChCl][EG] (a) and [ChCl][Urea] (b).

The CD spectra indicate that the peptide crosslinker adopts an α -helical conformation. This is in accordance with the fact that the peptide segments incorporated in the gels also adopt an α -helical conformation. Previously Gary et al also found that the secondary structure of proteins such as horseradish peroxidase and bovine serum albumen, could be maintained in certain DESs. (Journal of Materials Chemistry B, 2021, 9(3): 536-566) Comparison of the CD spectra of the peptide crosslinker and that of the gel reveals that the helicity of the peptide segments in the eutectogels is higher than the free peptide crosslinker dissolved in DES. Other authors also found immobilization of peptide chains enhances the stability of α -helix.

To acquire the CD spectra of the gel samples before stretching, being stretched at a certain strain and unloaded, we designed a stretching device which can fit into the sample cell of the circular dichroism spectrometer. The schematic diagram and the picture of the stretching device were added as Supplementary Figure 27 and also shown below:

Supplementary Figure 27 | The schematic diagram (a) and photograph of the stretching device (b) used to measure CD spectra of stretched gels.

Using the stretching device, the gel sample can be stretched precisely to a strain from 0 to 2000%, and its CD spectra at the strain can be measured. In this way, the CD spectra of the gel sample before stretching, being stretched at 1500% strain and unloaded were measured. In the revised manuscript, the part about CD measurement was revised to be: “Circular dichroism (CD) was obtained using a circular dichroism spectrometer (BioLogic, MOS-450) with 150 W Xenon lamp. A home-made stretching device which can fit into the sample cell of the circular dichroism spectrometer was used to stretch the gel sample to a predetermined strain. The schematic diagram and the picture of the stretching device were shown in Supplementary Figure 27.”

According to the reviewer’s suggestion, we strengthened the discussion about the CD spectra: “As revealed by CD spectra (Supplementary Figure 14) the peptide crosslinker dissolved in the DESs adopts an α -helical structure, which is evidenced by the two minima at 213 and 225 nm. More importantly the peptide chains incorporated in the eutectogels also adopt an α -helical structure. (Supplementary Figure 14 and 15) It is well-known that the α -helical structure is stabilized by hydrogen bonds between the C = O group of each amino acid and the NH group of amino acid four residues earlier in the sequence²⁷. These intramolecular hydrogen bonds can be fractured by external force, and thus dissipating energy applied on the gel²⁸. According to previous AFM study, the energy required to break the intramolecular hydrogen bonds is 20.2 kJ/mol²⁸. The breakage of the intramolecular hydrogen bonds leads to the destruction of the α -helical structure, which was evidenced by significant change in the CD spectra of the stretched gel (Supplementary Figure 15).”

Response to Reviewer #3's comments:

Reviewer #3 (Remarks to the Author):

The paper deals with the preparation of peptide-based polymeric eutectogels and their use as strain/pressure sensors.

The manuscript shows a high level of novelty for such kind of materials. Indeed, as stated by the AA, such kind of systems are frequently prepared in water. The manuscript is well written and properties as well as applications of the materials obtained are deeply analysed.

As a consequence, conclusions are well supported by experimental data.

For all above reasons, this referee thinks that the manuscript is suitable of publication in the present form.

Answer: We are very grateful for the reviewer's positive comments.

Response to Reviewer #4's comments:

Reviewer #4 (Remarks to the Author):

The manuscript by Zhang and co-workers details the preparation of polymer gels prepared from a deep eutectic solvent (DES) where an alpha-helical peptide is embedded into the structure. The resulting gel has impressive tensile and compressive strength, fracture toughness, fatigue resistance and adhesion. The authors show excellent repeatability with respect to sample deformation over 10000 cycles. The use of a deep eutectic solvent confers properties on the gel that are not normally accessible compared to those prepared in water, including use in sub-zero temperature conditions.. These materials have potential for use as conductive, flexible strain sensors.

The process is driven by a cross-linker that consists of a di-functional polypeptide consisting of polymerizable vinyl groups at each terminus, in the UV-mediated polymerization of an acrylamide matrix from two different Type 3 DES systems (choline chloride/urea and choline chloride/ethylene glycol). The resulting gels are highly transparent and stretchy, and can be used for 3D printing. The mechanical performance of these gels is significantly enhanced compared to a traditional cross linker (bis-acrylamide). The gels are tested in the context of strain sensors and pressure sensors with good performance. The gels are remarkably good at energy dissipation, which is attributed to the structure of the peptide cross-linker.

This is an impressive piece of work and the benchmarking of performance relevant to other reported eutectic-derived gels. Based on the Supporting Information this gel has the 2nd highest resilience, 4th highest toughness and 2nd highest fracture energy of the 9 reported. The elongation and strength at break are also very good but probably mid-range with respect to other DES-based systems; however importantly one does not trade off one category for another. The simplicity of synthesis, and convenience of using deep eutectic solvents makes this a notable contribution.

I have some specific points to be raised to the authors regarding questions that I feel are unresolved in the text and should be considered.

1. The peptide cross-linker (PC) has two vinyl groups of differing reactivity (an allyl terminus and an acrylate terminus). These are likely to have different reactivity ratios in the copolymerization of acrylamide which possesses an electron withdrawing group. Has this been confirmed - for example are there unreacted allyl pendants (for example) present within the final structure after photopolymerization?

Answer: Many thanks for the good question. As the reviewer pointed out, the peptide cross-linker synthesized here has two vinyl groups with different reactivity. The successful synthesis of gels using the PC as crosslinker demonstrated that both vinyl groups are active enough to participate in the polymerization process, leading to successful crosslinking of the products. To further examine if there is unreacted vinyl pendants present within the final structure, FTIR and ^{13}C solid-state NMR spectra of the gels (after washing out the DES) were measured. The results were added as Supplementary Figure 8 and 9 and also shown below:

Supplementary Figure 8| FTIR spectra of A_{0.4}PC22_{1%} and A_{0.4}BIS_{1%} gels. The gels were synthesized in [ChCl][EG]. Before measurement, the gels were washed with DI water to remove the DES and dried.

Supplementary Figure 9 | ^{13}C solid-state NMR spectra of $\text{A}_{0.4}\text{PC}_{221\%}$ (a) and $\text{A}_{0.4}\text{BIS}_{1\%}$ gels (b). The gels were synthesized in $[\text{ChCl}][\text{EG}]$. Before measurement, the gels were washed with DI water to remove the DES and dried.

One can see both the FTIR and ^{13}C solid-state NMR spectra of the PC-crosslinked gel were similar to that of the BIS-crosslinked gel. From the FTIR spectra, no signals of vinyl groups were found. Also no signals were found for vinyl groups on the ^{13}C solid-state NMR spectra. These results indicate that the amounts of unreacted double bonds left in the gels were negligible.

In the revised manuscript, a brief discussion about the results was added as follows in the revised manuscript: “FTIR and ^{13}C solid-state NMR studies revealed a similar chemical structure of peptide-crosslinked gels with the BIS-crosslinked ones. (Supplementary Figure 8 and 9) The amount of unreacted double bonds left in the gels was negligible.” (Page 6 Line 9)

2. Acrylamide itself has the potential to participate into the eutectic structure given its capacity to H-bond. Has this been examined, for example by 1D/2D NMR of the eutectic with acrylamide included? I anticipate that the authors have created a ternary eutectic (quaternary if you consider the PC).

Answer: Many thanks for the inspiring suggestion. According to the suggestion, we tried to test if acrylamide itself participates into the eutectic structure. Since we are not familiar with NMR technique, we adopted a method from a recent reference, in which the formation of ternary DES by adding water into urea/choline chloride DES was confirmed by measurement of the freezing point of the mixtures. (The Journal of Physical Chemistry B, 2019, 123(25): 5302-5306)

Following the reference, we prepared a series of AAm solutions in $[\text{ChCl}][\text{EG}]$. The solutions were placed in a cooling bath of ethanol/liquid nitrogen and the freezing points of the solutions were measured with a digital thermometer. The results were added as Supplementary Figure 7 in the revised manuscript and also shown below:

Supplementary Figure 7 | Freezing point of acrylamide solutions in [ChCl][EG] as a function of molar ratio of acrylamide/ChCl.

As the figure shows, adding acrylamide into [ChCl][EG] lowers its freezing point. The lowest freezing point was observed at a molar ratio of acrylamide/ChCl of 0.75. Compared to pure [ChCl][EG], the freezing point is decreased by about 41 °C at this molar ratio. Such a large drop in freezing point demonstrates that a ternary DES forms as the reviewer anticipated.

In the revised manuscript, along with the adding of Supplementary Figure 7, the following sentence was added on Page 6 Line 3: “Adding AAm actually forms a ternary DES as evidenced by further decrease in freezing point. (Supplementary Figure 7)”

3. In the text (lines 159-168), the authors refer to changing the PC loading, but are referencing the wrong figures (this should be Figs 3e /3f).

Answer: Many thanks for pointing out this problem. To correct this problem, Fig 3 was redrawn. The order of Figures 3c/3d and 3e/3f was changed. The caption was modified accordingly.

4. Figure 5 - what is the mass of the eutectogel in panel a? It would be interesting to know the mass/mass ratio that these gels can support via adhesion

Answer: Many thanks for the question. The size of the gels used in panel a of Fig.5 were 10 mm × 10 mm × 2 mm. The mass of the gels used was approximately 0.27 g. The gel can stick and lift items with a mass over 100g. So the mass/mass ratio of the gel and items they can support via adhesion is over 370. In the revised manuscript, the following sentence was added in the caption of Fig. 5: “The mass of the gels was ~0.27 g.”

5. In all photos of gels (e.g. Fig 6f and similar), please provide a scale bar for reference.

Answer: Many thanks for the suggestion. According to the suggestion, Fig. 2, 4, 6 and 7 were redrawn. Scale bars were added in Fig. 2b, 2e, 4a, 6a, 6f and 7a .

6. I am confused by the DSC trace in Figure 6c. Firstly, is this a heating or cooling trace? Secondly, provide the trace in the opposite direction. Were multiple heating/cooling loops performed? An endothermic transition is visible, can the authors describe this (is it a melt transition?) DSC hysteresis is often seen in deep eutectic solvents and is worth discussing. The authors discuss that the gel freezing point is -54.4 deg C (line 284), however this temperature is not even in the scale of the figure.

Answer: Many thanks for pointing out this problem. The DSC trace in the original version is a heating trace without pre-hand heating/cooling loops. According to the reviewer's suggestion, we measured the DSC trace again. A wide temperature range was used. Before measurement, the thermal history was first removed by performing two heating/cooling cycles. Both heating and cooling traces were measured. The results were shown in the figure below:

DSC curve of an $A_{0.4}PC_{22}1\%$ [ChCl][EG] gel.

Unlike the DSC thermogram in the original manuscript, no transition was observed at 100°C from the new DSC thermogram, likely because the thermal history was removed before the measurement. The freezing point was determined to be -51.2°C (liquidus temperature in the heating process). As the reviewer anticipated, a hysteresis observed: while the liquidus temperature was determined to be -51.2°C in the heating process, it shifts to -60.9°C in the cooling process. The hysteresis may be ascribed to the high viscosity of the system.

In the revised manuscript, Fig. 6c was replaced with the new DSC results. The sentence “DSC study further confirms that the freezing point of the gel is -54.4°C , lower than the freezing point of pure [ChCl][EG] (-36.3°C)⁴⁵ (Fig. 6c).” was corrected to be “DSC study further confirms that the freezing point of the gel is -51.2°C , lower than the freezing point of pure [ChCl][EG] (-36.3°C)⁴⁵ (Fig. 6c).” (Page 14 Line 9)

7. One thing standing out for me is that these eutectogels can potentially be purified (the EG/urea and choline chloride can be removed via washing) and the resulting cross-linked network can be re-dispersed/hydrated with water. Based on the results presented, this gel should show inferior performance. It would however be an interesting reference point in comparison to the gels prepared

directly in water (e.g. the data in Figure 5b). I raise this, as the high viscosity of the eutectic changes the nature of the photopolymerization process and thus cross-linking density and ultimately mechanical response.

Answer: Many thanks for the inspirational suggestion. As the reviewer pointed out, the high viscosity of the deep eutectic solvent could influence the photopolymerization process and thus cross-linking density and ultimately mechanical properties of the gel. According to the reviewer's suggestion, we tried to change the solvent of eutectogels from [ChCl][EG] to water and compare their properties with hydrogels directly prepared in water. However, one should keep the size of the gel unchanged after the solvent change, which demonstrated highly difficult. We finally found a method to do so using a mould assembled from ceramic foam, which is porous and permeable. Using the mould, eutectogels were synthesized and then the solvent was changed in situ by soaking the mould in a series of mixed solution of [ChCl][EG] and CaCl_2 aqueous solution. Finally the solvent of the gel was successfully changed to be CaCl_2 aqueous solution, but the size of the gel remained unchanged. The following shows the picture of the gel sample (left) and another gel synthesized directly using CaCl_2 aqueous solution as solvent in the same mould (right):

The mechanical properties of the two gels were then measured. As shown below, compared with the gel synthesized directly in aqueous solution, the gel synthesized in DES exhibits a higher elastic modulus and lower elongation at break:

The result confirms the hypothesis proposed by the reviewer: the high viscosity of the deep eutectic solvent could influence the photopolymerization process and thus cross-linking density and ultimately mechanical properties of the gel. Specifically, the high viscosity of the deep eutectic solvent leads to a higher cross-linking density of the gel, and therefore higher elastic modulus and lower elongation at break.

Reviewers' Comments:

Reviewer #2:

Remarks to the Author:

The authors have adequately addressed all the concerns and comments raised during the first round of revisions. I acknowledge the number of new experiments that helped improve the discussions. The manuscript should be ready for publication in its present form.

As for the comments and suggestions made by Referee 1, all were addressed point-by-point with a solid rebuttal based on detailed comparisons with the properties exhibited by similar materials prepared with water. It was clear that other hydrogel-based materials outperformed the specific properties of the eutectogel described here. However, the peptide-containing eutectogel stands out with overall performance, i.e., simultaneously shows very competitive toughness, resilience, adhesion, anti-freezing, and anti-drying properties. I agree with Referee 1 about the comment on the preliminary biocompatibility test, but the authors have deleted the part accordingly. Therefore, I ratify my decision to recommend the manuscript for publication.

Josué D. Mota-Morales

Reviewer #4:

Remarks to the Author:

The revised manuscript by Zhang et al details the preparation of polymeric gels from polymerizable deep eutectic solvents using a peptide-based cross-linker that have excellent properties as conductive strain/pressure sensors.

The initial manuscript was assessed by four reviewers and a comprehensive battery of changes have been made to the revised manuscript. This includes significant numbers of new experiments as requested by myself and other reviewers of the initial manuscript. In my opinion these clearly address the concerns I had and I believe the response to other reviewers has been performed admirably as well.

The suite of impressive resilience/toughness/strength/conductivity make these an appealing class of materials. I feel that this manuscript will be of high interest and relevance to many in the readership of this journal.

Response to Reviewer #2' comments:

The authors have adequately addressed all the concerns and comments raised during the first round of revisions. I acknowledge the number of new experiments that helped improve the discussions. The manuscript should be ready for publication in its present form.

As for the comments and suggestions made by Referee 1, all were addressed point-by-point with a solid rebuttal based on detailed comparisons with the properties exhibited by similar materials prepared with water. It was clear that other hydrogel-based materials outperformed the specific properties of the eutectogel described here. However, the peptide-containing eutectogel stands out with overall performance, i.e., simultaneously shows very competitive toughness, resilience, adhesion, anti-freezing, and anti-drying properties. I agree with Referee 1 about the comment on the preliminary biocompatibility test, but the authors have deleted the part accordingly. Therefore, I ratify my decision to recommend the manuscript for publication.

Answer: We are very grateful for the reviewer' s positive comments.

Response to Reviewer #4' comments:

The revised manuscript by Zhang et al details the preparation of polymeric gels from polymerizable deep eutectic solvents using a peptide-based cross-linker that have excellent properties as conductive strain/pressure sensors.

The initial manuscript was assessed by four reviewers and a comprehensive battery of changes have been made to the revised manuscript. This includes significant numbers of new experiments as requested by myself and other reviewers of the initial manuscript. In my opinion these clearly address the concerns I had and I beleive the response to other reviewers has been performed admirably as well.

The suite of impressive resilience/toughness/strength/conductivity make these an appealing class of materials. I feel that this manuscript will be of high interest and relevance to many in the readership of this journal.

Answer: We are very grateful for the reviewer' s positive comments.